# On the Self-Verification Limitations of Large Language Models on Reasoning and Planning Tasks

**Kaya Stechly**[*]
Arizona State University
khstechl@asu.edu

**Karthik Valmeekam**[*]
Arizona State University
kvalmeek@asu.edu

**Subbarao Kambhampati**
Arizona State University
rao@asu.edu

## Abstract

There has been considerable divergence of opinion on the reasoning abilities of Large Language Models (LLMs). While the initial optimism that reasoning might emerge automatically with scale has been tempered thanks to a slew of counterexamples–ranging from multiplication to simple planning–there persists a widespread belief that LLMs can, in a general fashion, self-critique and improve their own solutions iteratively. This belief seemingly rests on the assumption that verification of correctness should be easier than generation–a rather classical argument from computational complexity–which should be irrelevant to LLMs to the extent that what they are doing is approximate retrieval. In this paper, we set out to systematically investigate the effectiveness of iterative prompting in the context of reasoning and planning. We present a principled empirical study of the performance of GPT-4 in three domains: Game of 24, Graph Coloring, and STRIPS planning. We experiment both with the model critiquing its own answers and with an external correct reasoner verifying proposed solutions. In each case, we analyze whether the content of criticisms actually affects bottom line performance, and whether we can ablate elements of the augmented system without losing performance. Across most of our domains, we observe significant performance collapse with self-critique and significant performance gains with sound external verification. We also note that merely re-prompting with a sound verifier maintains most of the benefits of more involved setups.

## 1 Introduction

Large Language Models (LLMs) have rapidly captured the attention of the AI research community with their exceptional natural language completion capabilities. Trained on web-scale language corpora, these models have demonstrated the ability to generate seemingly valuable completions across a wide range of topics. This has led to a surge of interest in determining whether such models are able to perform well on reasoning tasks. Though initial anecdotal results showed promise, further systematic studies revealed inconsistencies and significant issues when applied to reasoning tasks–such as simple arithmetic or logic (Dziri et al., 2023) and planning (Valmeekam et al., 2023b). These results have questioned the robustness of their reasoning abilities and led researchers to explore ways to augment and improve these systems.

Of particular interest to us are emerging claims about LLM self-critiquing. In a typical setup, an LLM is iteratively prompted so that it both generates candidate solutions and, in response to separate queries, evaluates them. This process is looped until some stopping condition is met in hopes of potentially generating a refined answer. Current works (Yao et al., 2023a; Shinn et al., 2023; Weng et al., 2023; Chen et al.; Madaan et al., 2023), while admitting that LLMs are not good reasoners on their own, still exhibit considerable optimism about such self-critique systems. This belief seems to rest largely on the assumption that verification of correctness should be easier than generation for many reasoning problems–a rather classical argument from computational complexity. However, we think there are grounds to be skeptical of this assumption. The complexity of the reasoning task

---

[*]Equal Contribution

should be largely irrelevant to LLM performance, especially if what they are doing is better modeled as approximate retrieval (Kambhampati, 2024).

Intrigued by the prevailing optimism, in this paper we set out to systematically investigate the effectiveness of using LLMs to critique their own generations in the context of planning and reasoning. To gain meaningful insights into the verification/critiquing abilities of LLMs for reasoning tasks, it is crucial to test them on *formal* tasks–ones where machine-verifiable ground truths are available for both generation and criticism. Note that such verification is not feasible in style-based/qualitative tasks like creative writing (Yao et al., 2023a) where there is no formal notion of correctness and critique can vary widely. With this understanding, we select three distinct reasoning problems (*Game of 24*, *Graph Coloring*, and *STRIPS planning*), in which there exist formal notions of correctness that allow us to automatically check the veracity and quality of both (binary) verification and critique generated by the LLM. Furthermore, these domains are chosen so that the algorithmic abilities they test are fundamental—any other reasoning task must include components that test these same capabilities, or else be only a retrieval task.

Our methodology employs a system (which we refer to as LLM+LLM) that uses the same LLM (GPT-4-0613 (Achiam et al., 2023)) for both solution and feedback generation. A generation prompt is sent to the LLM. Its response is used to create a verification prompt, which is then sent back to the same LLM. We use the feedback generated in this way to then create a backprompt, thus restarting the cycle.

Across most of our domains, this self-verification system *worsens* performance. We find that as the number of backprompts increases, this kind of self-correction consistently degrades output quality. Our analysis reveals that the verifier LLM's false negative rate is significant across our domains. In essence, even when the LLM generates a valid solution, the verifier LLM rejects it often enough that overall performance suffers.

We contrast this performance collapse with two baselines. The first is an ablated variant of the system (which we refer to as LLM+Sound Verifier), where an external sound verifier evaluates the LLM's generations and produces critique. This setup gives substantial performance gains across all domains, but closer analysis shows that the level of feedback often doesn't seem to matter–as long as the verifier is sound, improvement remains regardless of how much or how little feedback the LLM receives.

We ablate the system further, and remove critique entirely. In this setup, the LLM is repeatedly queried with the exact same base prompt until a sound verifier certifies its solution. Within this impoverished setting, prompts no longer maintain a past history of guesses, yet we can maintain most, if not all, of the gains shown by our previous, more complicated, more expensive setups.

Our empirical results suggest that the benefits of iterative prompting and verification can easily be misattributed to opaque self-critique and seemingly rich feedback. Thus, future implementations of LLMs for reasoning tasks should take the form of LLM-Modulo systems (Kambhampati et al., b) where verification is done by external sound systems. In the rest of the paper, we first review related work and discuss domain backgrounds. Then, we explain our methodology, and finally closely analyze LLM self-verification abilities on our domains.

## 2 RELATED WORK

Following the release of GPT-4, anecdotal accounts of performance on reasoning tasks (Bubeck et al., 2023) spurred much research into the capabilities of LLMs on various reasoning domains, from a menagerie of benchmarks covering basic problems (Kojima et al., 2022) to planning (Valmeekam et al., 2023b), logic and arithmetic (Dziri et al., 2023), analogical reasoning (Webb et al., 2023), and even math puzzles (Yao et al., 2023a). Though these seemed initially promising, systematic studies began to generate negative results across many of these domains (Valmeekam et al., 2023b; Silver et al., 2022; Abdin et al.; Ullman, 2023; Gendron et al., 2024), claiming that LLM scaling shows much lower returns for reasoning tasks (Rae et al., 2021), and showcasing brittle performance in the face of minor problem permutations (McCoy et al., 2023; Dziri et al., 2023; Arakelyan et al., 2023).

In response, researchers created augmented systems which embed the LLM in larger frameworks in an attempt to improve performance. These take many forms: common search algorithms with the LLM cast in the role of heuristic (Yao et al., 2023a; Hao et al., 2023), approaches which reduce error rates by enforcing various consistency conditions (Du et al.; Cohen et al., 2023; Jiang et al.,

2023), and direct LLM self-critique (Shinn et al., 2023; Weng et al., 2023; Chen et al.; Huang et al., b; Madaan et al., 2023; Yao et al., 2023b).

In the current work, we are interested in examining this third approach: LLM self-critique. In the most basic case, the LLM is queried for an answer, and then is re-queried with its own response together with some instructions to critique or improve it, with this process looped until some stopping condition. This is fundamentally based on the intuition that verification is easier than, or at least different enough from, generation that such a process can improve performance–in analogy to human self-critique(Weng et al., 2023).

The literature abounds with strong, well-cited, and well-referenced claims about the efficacy of these techniques. (Shinn et al., 2023) claims there is an "emergent property of self-reflection in LLMs" and that "self-reflection is extremely useful to learn complex tasks over a handful of trials." Their experiments claim that every variety they try leads to improvement, and that this is because "self-reflective feedback acts as a 'semantic' gradient signal by providing the agent with a concrete direction to improve upon, helping it learn from prior mistakes to perform better on the task."[1] Other works claim this self-correction does not require "any human feedback" (Chen et al.) and that "even when an LLM cannot generate an optimal output on its first try, the LLM can often provide useful feedback and improve its own output accordingly," (Madaan et al., 2023) seeming to indicate that these claims generalize beyond the domains, problems, and prompts they were originally made for.

However, some further systematic investigations have found less impressive results in logical fallacy detection (Hong et al., 2024) and HotpotQA (Huang et al., a), demonstrating very brittle improvement at best, some of which could be replicated sans self critique by merely including missing domain-general information into the original prompt. The authors of the CRITIC framework(Gou et al.) were the first to notice that, in some cases, LLM self-critique can lead to decreases in performance when compared to sound verification. Contemporaneous to our work,[2] (Huang et al., a) investigate two-round self-correction schemes in the GSM8K, CommonSenseQA, and HotpotQA domains. They compare which answers were changed (from correct to incorrect or incorrect to incorrect) and which weren't, and discuss extensions of their argument to multiagent debate.

Our own work focuses on autonomous multi-round self-verification within three formally verifiable domains that reflect reasoning tasks. We extend previous work by ablating the self-critique system thoroughly to pinpoint the source of performance deterioration, considering more prompting rounds (up to 15), and by examining a new set of domains which we argue are better and more broadly applicable tests of reasoning and self-correction capability.

Reasoning is a fraught term. Previous work has used it to refer, among others, to the human ability to draw conclusions(Leighton & Sternberg, 2004), to the ability to apply common sense to simple scenarios, to positive performance on short-form written tasks, and to formal deductive inference. However, it is often unclear which definition a given set of authors presupposes when making claims about LLM reasoning capabilities. This muddies the discussion and contributes to a strange duality: highly cited papers claim that LLMs are general-purpose reasoners (Kojima et al., 2022; Wei et al., 2022; Bubeck et al., 2023; Zhou et al.); that they have strong, human-like self-reflection capabilities which allow them to correct reasoning mistakes they do make (Shinn et al., 2023; Chen et al.; Madaan et al., 2023); that they can answer difficult, never-before-seen questions via in-context learning as long as they are allowed to use chain of thought to generate intermediate scratch work (Dong et al., 2024); that they can pass or come close on many high school and college-level examinations (Achiam et al., 2023; Gilson et al., 2023; Raimondi et al., 2023; Thaker et al., 2024; Yeadon & Halliday, 2023; de Winter, 2023; Kortemeyer, 2023) and that performance on such standardized exams is evidence about their reasoning capabilities and domain expert knowledge (Wang et al., a). Yet, responses to counterexamples and negative results, anecdotally, fall back on a much weaker, seemingly contradictory constellation of premises: LLMs only perform well on things they were trained on, and–in fact–if a model performs poorly, we can only conclude it wasn't trained on that (but if it performs well, it is generalizing); the average non-expert human would fail on this task if presented it with zero context or training, therefore it's unsurprising that the LLM fails; no good

---

[1]Note that our results ablate away much of this signal (especially the 'concrete direction' that exists in explicit critique) to find that most of the improvement in our domains comes from the soundness of the verifier.

[2]Preliminary results from our work were originally presented in two papers at a NeurIPS 2023 workshop.

prompt engineer would query the LLM in this fashion. (How to tell if a prompt is good? It follows one of several anthropomorphized design patterns and, most importantly, the result is positive.)

These shifting definitions and implicit assumptions make it very difficult to make concrete claims and expect to be understood, and they make it even more difficult to pin down claims made by others or attempt to falsify them. In the current work we address this by restricting our focus to fully specified, formally verifiable problems which can be solved directly by deductive methods. Though these may at first seem like a very narrow class, especially when compared to the cornucopia of commonsense, language-based, and domain-specific benchmarks in the literature, we argue that they are fundamental, as any other reasoning tasks must include components that test these same capabilities–otherwise they are merely testing recall. Our work extends studies that have looked at similar problems, especially those that examined LLM planning capabilities and other classical reasoning problems (Valmeekam et al., 2023a;b; Stechly et al., 2024; Dziri et al., 2023). However, no previous work has looked carefully at a broad range of formal verification problems. Filling in this gap is important, as a lack of benchmark coverage contributes to the illusion that LLMs possess greater competency than they really do (Saxon, 2024).

Furthermore, common domains fall short for evaluating the reasoning and self-critique abilities of LLMs for additional reasons: test set memorization, lack of problem difficulty, and lack of ground truth.

**Test set memorization:** Due to the black box nature of state of the art models, ensuring that they weren't trained on those problems is difficult, and there is compelling evidence that they have memorized significant chunks of common benchmark sets (Roberts et al., 2023). Many benchmark sets do not allow for arbitrary generation of novel questions, or worse, draw data from publicly available sources–the same sources LLM trainers have access to (Yang et al., 2018; Srivastava et al., 2023). We consider arbitrary generation of new instances of varying difficulty a key desideratum for any evaluation domain.

**Lack of problem difficulty:** Some of the benchmarks (e.g. HotPotQA (Huang et al., a), GSM-8k (Madaan et al., 2023))used in evaluations of self-verification are easy–that is, SoTA LLM performance is already high–and are therefore much less informative about the effects of the refinement procedure, especially if we are interested in how much generalization this procedure allows beyond baseline direct prompting. Additionally, many such sets over-constrain the solution space, usually by putting the question into multiple choice format. Not only does this make valuable and interesting critique hard to produce and evaluate, but it trivializes refinement: even a very simple agent can solve an $n$-choice problem with $n - 1$ critiques–just don't repeat the same answer. Conclusions drawn over reduced problem spaces of this type are unlikely to generalize.

**Lack of ground truth:** A number of tasks that LLMs are evaluated on (e.g. writing prompts (Yao et al., 2023a), constrained text generation (Lin et al., 2020), toxicity control (Welleck et al., 2022; Gou et al.) , etc.) are problems without a well-defined ground truth. Instead, they are evaluated by a couple of indirect methods. Some require an assorted set of metrics which may not be well-validated for LLMs (e.g. see Ullman (2023) for discussion on problems with transferring results from human-validated tests). Some are scored by humans (Yao et al., 2023a). And some are evaluated by another pre-trained language model (Madaan et al., 2023) or black box AI evaluator (Welleck et al., 2022). This makes conclusions much harder to draw.

## 3 BACKGROUND ON TEST TASKS

We evaluate GPT-4's self-critique abilities over three distinct tasks, chosen because we believe they are good proxies for harder reasoning tasks, and because they allow freedom in arbitrary generation of additional instances while providing easy-to-deploy formal verifiability and guaranteed quality.

This gives more than just flexibility–it also decreases the chance that our instances are represented in the black box model's opaque training sets. This strengthens our results by reducing the likelihood that the model can substitute approximate retrieval for more general reasoning ability.

Any given problem in our sets also has the property that it has a large number of potential solutions, and this solution space cannot be substantially reduced through simple pattern-matching. As we are interested in self-verification loops where the LLM has access to its previous guesses, it is very

important that removing a handful of possible solutions does not trivialize the problem. Compare this to common multiple choice question datasets, where any $n$-option problem can be solved in $n$ exclusive guesses.

## 3.1 GAME OF 24

*Game of 24* is a math puzzle where the goal is to combine four numbers with parentheses and basic arithmetical operations (addition, multiplication, subtraction, and division) to create an expression that evaluates to 24. The numbers are typically constrained to the range 1-12, a nod to game's playing card roots. Previously, it has been used as a domain of evaluation for other LLM self-verification schemes ((Yao et al., 2023a) and fulfills our domain desiderata (see 2). We use it here to enable direct comparisons between previous work and the current paper.

Following (Yao et al., 2023a), we use data scraped from `4nums.com`. This list of problems is ordered from shortest to longest average human solution time. Like (Yao et al., 2023a), we evaluate our generation tasks on instances 901-1000. However, when evaluating verification and critique alone, we use instances 1-1000.

Verification in this domain is straightforward: given a proposed expression, simplify it using basic arithmetic and check if it is equal to 24. As a sound verifier, we use `SymPy`[3], a common Python library for symbolic mathematics, and handle any errors that it throws (for instance, if there are unbalanced parentheses) by outputting feedback that says the LLM's generation was malformed.

## 3.2 GRAPH COLORING

*Graph coloring* is a a canonical NP-complete reasoning problem that is related to both propositional satisfiability as well as practical problems like scheduling and allocation. The complexity class NP contains problems that are hard to solve, but easy to verify, so this allows our It is broad enough to give insights into reasoning more generally, yet simple enough that it can be specified and evaluated by a human or basic pattern matching.

In this work, an instance of a graph coloring problem consists of a planar graph together with an optimal coloring number $n$. The goal is to output a solution that assigns one of $n$ colors to each vertex such that no two edge-connected vertices share a color.

Using `GrinPy`[4] to handle common graph operations, we built a test set of 100 graphs of varying small sizes. Each graph was constructed using a variant of the Erdős–Rényi method ($p = 0.4$), with any non-planar or repeat graphs discarded. These were compiled into the standard DIMACS format (DIMACS) together with the graph's precalculated chromatic number.

Verifying that a proposed coloring is correct is also easy: just check the colors of every edge. If any of them has two vertices of the same color, reject the coloring. Our sound verifier is a simple, single `for`-loop implementation of this idea in Python: for each edge in the graph description, we check that both of its vertices are different.

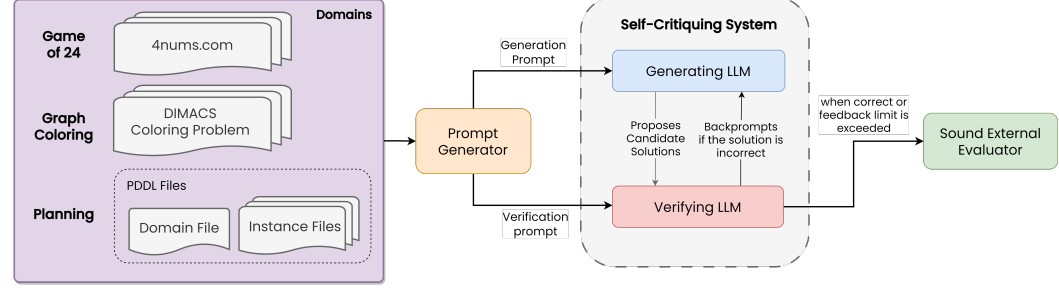

Figure 1: **Overall Backprompting and Evaluation Architecture**

---

[3]https://www.sympy.org/en/index.html
[4]https://pypi.org/project/grinpy/

### 3.3 STRIPS PLANNING

STRIPS planning is a formalism used in automated planning that operates in a discrete, deterministic space. Such problems, commonly known as classical planning problems, involve finding a sequence of actions that when executed in a certain world state will take the agent to a desired goal state. STRIPS planning has a long history of featuring in various planning challenges and competitions, and is formally represented using the PDDL (Planning Domain and Definition Language) framework (McDermott et al., 1998). We consider two domains in STRIPS planning. One is *Blocksworld*, a simple common-sense domain used in International Planning Competitions (IPC, 1998) and *Mystery Blocksworld*, which is an obfuscated version of Blocksworld. For both the domains, we draw instances from (Valmeekam et al., 2023a) for our evaluations.

A PDDL specification consists of three components. The *domain* doesn't change between problems and consists of a set of predicates, which can be used to describe the state of the world, and a set of actions–stored with their preconditions and effects–that the agent is allowed to take. The *initial state* is a list of predicates that are true at the outset of the specific problem (an example predicate, in natural language: "the red block is on the table"). The *goal* is a boolean expression of predicates.

Solutions to PDDL problems take the form of correct plans–sequences of actions which can be executed from the initial state without violating any of their preconditions and which arrive at a final state that satisfies the goal. Verifying proposed plans is a matter of following the actions in order and checking that these two desiderata are achieved. For our experiments, we use VAL (Howey et al., 2004) as the sound external critique that evaluates and critiques LLM generated plans.

## 4 METHODOLOGY

As our results are about ablations of self-critique architectures, our basic test framework is a general prompting architecture informed by interchangeable domain-specific components. Our intent is to give the LLM as much information as possible, so we design prompts that include the entire history of previously proposed solutions and the feedback they received.

A problem instance is a domain-specific formal description. In attempting a problem, our system (as shown in Figure 1) proceeds as follows: (1) The instance is processed by a simple, hard-coded natural language translator into a prompt and sent to the LLM. (2) The LLM's response is wrapped in a domain-specific critique prompt, which is separately sent as another LLM query. (3) If the following response claims that the proposed solution is correct, we stop the system and output the proposed solution. If it doesn't, the critique is extracted, wrapped in instruction text, and appended to a prompt containing the entire history of interactions so far. This is then sent to the LLM, and the cycle repeats, potentially until we enforce a timeout.

Though only two types of prompts are sent, the LLM can be seen as playing three separate roles: as an answer guesser, a (binary) verifier, and a critique generator. In order to better understand which of these roles contribute to increased performance, we will examine variations of this system where one or more of them are changed or removed.

To examine LLM verification abilities, we first measure the performance of the entire system, and then evaluate false positive and false negative rates across domains. To better understand the guesser role, and the LLM's ability to consider and implement critique, we will modify the loop so that the verification and critique roles are played by a provably sound verifier that provides rich, correct feedback. We will then reduce and eventually eliminate the amount of provided information (e.g. rich feedback: explicitly giving an evaluation of a proposed Game of 24 expression; minimal feedback: "the previous answer was wrong"; no feedback: re-querying with the base prompt), while keeping track of changes in the performance of the entire system.

For LLM critique generation, we construct subdomains of our original domains. In these prompts, we provide a problem description and a proposed solution, and we ask the LLM to provide domain-specific critique of the solution if it is incorrect. We parse the output using a hard-coded script and measure accuracy compared to the sound verifier's output.

Note that sound verifiers output task specific critiques: for Game of 24, the evaluation of the provided expression ("1+1+4+6=12 not 24"); constraint violations for graph coloring ("vertices 1 and 3 were

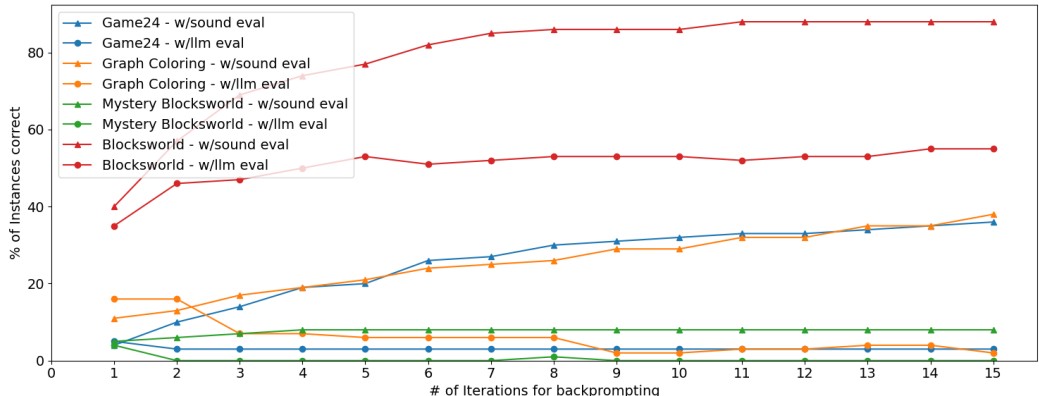

Figure 2: **Performance vs Number of Iterations Before Timeout.** We measure performance at iteration $n$ by evaluating how many instances are currently correctly solved (whether the LLM has verified them or not. In other words, we evaluate as if the timeout were $n$) and adding that to the number the LLM has correctly verified so far. Note that if the verifier incorrectly rejects an answer and the followup is wrong, the next round's percent correct may decrease. When paired with a sound verifier, the LLM monotonically improves its performance as the number of backprompts increase up to some asymptote. The top three lines show this for each of our domains. On the other hand, when the LLM itself is used as a verifier, performance collapses immediately.

both colored red despite sharing an edge"); and precondition violations (the second action "succumb object a" is invalid because the succumb action requires the pain object to be true, which is not the case after the first action.) and failure to reach goal ("this plan does not reach the goal") for planning.

## 5    EXAMINING SELF-VERIFICATION

| Domain | S.P. | LLM+LLM | LLM+Sound Critique | | | Sampling | | S.C. |
|---|---|---|---|---|---|---|---|---|
| | | | B.F. | F.E.F | A.E.F | k=15 | k=25 | k=15 |
| Game of 24 | 5% | 3% | 36% | 38% | N/A | 28% | 42% | 6% |
| Graph Coloring | 16% | 2% | 38% | 37% | 34% | 40% | 44% | 14% |
| Blocksworld | 40% | 55% | 60% | 87% | 83% | 68% | 72% | 42% |
| Mystery Blocksworld | 4% | 0% | 10% | 8% | 6% | 9% | 14% | 4% |

Table 1: **Accuracy across prompting schemes over 100 instances per domain.** S.P.-Standard Prompting. B.F.-Binary Feedback. F.E.F-First Error Feedback, e.g. the first wrong edge, the first mistaken action, or the non-24 evaluation of the proposed expression. A.E.F-All Error Feedback, e.g. every wrong edge, every mistaken action and error. Note that there is no third critique type for Game of 24 due to the simplicity of the domain. We include two examples of sampling, one at 15 samples, the other at 25, to show that completely ablating critique retains the performance increases of critique. We also include S.C.-Self Consistency results, where the most common answer in a pool of 15 is the one that is output by the model, as another comparison point.

We evaluate our system over 100 instances in each domain. In standard prompting we send a single query to the LLM and treat whatever it outputs as its final answer. We use this as our baseline. As shown in Table 1, when we augment this condition with the full self-critique setup, performance *decreases*. In fact, Figure 2 shows that as the number of backprompts increases, this kind of self-correction consistently degrades output quality. If the LLM were a good verifier, then we would have

expected it to recognize instances which are already right, and thus–at worst–maintain the baseline score.

The LLM-as-verifier ranges in accuracy depending on the domain, as illustrated in Table 2. Notably, Game of 24 and Blocksworld maintain lower rates of both false positives and false negatives, and this is reflected in LLM+LLM performance on those domains, which doesn't fail as drastically as it does in cases. In Blocksworld, we even see a modest improvement, though that improvement is still significantly worse than having a sound verifier. In the remaining two domains, the false negative rates are very high. In effect, the system rejects most answers and then times out on a set of later, worse generations.

When we replace the LLM verifier with a sound verifier, every correct answer will be accepted properly. Intuitively, it can do no worse than standard prompting—anything that was generated correctly initially must be accepted. As shown in Table 1, performance is much higher in all sound critique cases, though it still falls short of 100%. Due to the setup, this can't be due to the verifier, but must be the fault of the answer generating LLM. After 15 rounds, any instance that has yet to been answered correctly will time out, and this process is the *only* source of inaccuracy arising from the LLM-sound verifier loop.

In general, it is clear that the verifier requires high accuracy or else the overall system will encounter compounding errors. In the reasoning domains considered, LLMs-as-verifiers are mostly insufficient to prevent these.[5]

| Domain | Accuracy | F.P.R | F.N.R |
|---|---|---|---|
| Game of 24 | 87.0% (3567/4100) | 10.4% (320/3071) | 20.7% (213/1029) |
| Graph Coloring | 72.4% (362/500) | 6.5% (25/382) | 95.8% (113/118) |
| Blocksworld | 71.8% (359/500) | 18.55% (64/345) | 15.48% (24/155) |
| Mystery Blocksworld | 79.6% (398/500) | 0.5% (2/397) | 97.09% (100/103) |

Table 2: **LLM Verification results.** F.P.R. - False Positive Rate, F.N.R - False Negative Rate.

### 5.1 CRITIQUE GENERATION

We consider the quality of LLM-generated free-form critiques separately from that of their binary verification signals, and find that they are full of unhelpful hallucinations and mistakes. To obtain the following results, we ran a further suite of experiments with specially crafted proposed solutions wrapped in verification prompts. The exact breakdown of which types of solutions were generated is available in each domain's appendix.

In Game of 24, without any further instructions, the LLM tends to output incorrect suggestions for the answer. When prompted to give an evaluation of the proposed expression first, its accuracy varies. In fact, when we restrict ourselves to only looking at verification of equations that are guaranteed to equal 24, and therefore must be correct, it labels $79.1\%$ of them as correct, but evaluates $81.6\%$ of them to 24. That is, there are problems which it will evaluate correctly but still mark wrong.

In Graph Coloring, the LLM's critiques of proposed solutions are riddled with non-existent edges and include many spurious claims about the colors of vertices, often missing the violated constraint in favor of them. A breakdown and detailed examples are provided in appendix A.4.1.

In the planning domains, the critiquing LLM often hallucinates whether action preconditions are met or not. In Mystery Blocksworld, the LLM incorrectly assumes the state of these preconditions as well. This leads to lower accuracy of the critiques provided by the LLM. A further breakdown is in appendix A.5.1

In other words, the LLM introduces errors in two places: verification, where it can pass over correct answers and accept wrong ones; and critique generation, where it can produce misleading feedback and bias future outputs away from the correct answer. When they compound sufficiently, these errors

---

[5]Prompt and response examples can be found in the Appendix.

actually reduce the performance of the LLM-based self-critique loop below that of just taking the LLM's very first guess.

## 5.2 CRITIQUE CONSIDERATION

Our results also imply that the generator LLM often isn't sensitive to varying levels of feedback. We use a sound verifier to critique the LLM's output, and compare the results over three levels of feedback for graph coloring and planning, and over two levels for game of 24. Examples of prompts containing each sort of feedback can be found in appendices A.3.3, A.4.3, and A.5.7.

- **Binary feedback** is the same for all domains: either the verifier accepts the solution, stopping the system, or we create a backprompt which says the previous answer was wrong but doesn't elaborate.

- **First error feedback** mentions the first error that was found when evaluating the response from top to bottom (e.g. an incorrect edge in graph coloring, an inexecutable step in planning, the evaluation of the proposed expression in game of 24).

- **All errors feedback** includes every error that was found. Note that due to the simplicity of game of 24, we do not implement a third feedback level for it.

Perhaps surprisingly, Table 1 shows very little difference between these three conditions. And in two of our domains, increasing the amount of feedback actually leads to a decrease in performance.

The intuition underlying the entire critique system is that sending a history of previous failed attempts together with information about why those attempts were failures should guide the LLM toward better future answers. If this were true, then we would expect the performance jump to be tied to the content of the backprompts. With only the data discussed so far, it might seem like the relevant content is actually the history of failed attempts the LLM receives rather than any feedback on those attempts. However, for our domains, our final experiments contradict this interpretation as well.

We take our ablation of critique consideration to the logical extreme, and remove the availability of critique entirely. In this sampling setup, we keep the verifier but don't change the prompt at all between iterations. The LLM (at $t = 1$) is asked the same question over and over until the verifier certifies it or it hits some pre-established timeout.

Represented in Table 1 by the "Sampling" columns, this gives comparable gains to feedback conditions. Note that, because prompts do not grow additively with iteration number, the token cost of these prompts is quadratically lower. This allows us to increase performance further by just increasing $k$ further. As a sanity check, we compare this to a self-consistency baseline (Wang et al., b), where we instead select the most common answer from the 15 generated ones. This baseline is listed under "S.C." and shows no improvement over standard prompting.

Our final results show that, in our domains, the information in critiques does not have as much of an effect on performance as previous literature claimed it should. In fact, our performance increases seem to stem in large part just from having enough guesses and a sound verifier. In reasoning problems we therefore cast the LLM primarily as an idea generator.

## 6 CONCLUSION

In this paper, we conducted a systematic evaluation of the self-critique abilities of Large Language Models on three reasoning and planning tasks. We separated self-critique into three components: verification, critique generation, and critique consideration. Across the hard reasoning domains we evaluated, LLMs did poorly in all three roles, with compounding errors from across these roles often making the LLM self-critiquing loop perform worse than just having the LLM guess the solution up front. These failures of verification could potentially be very detrimental to a system's dependability, especially in domains where high reliability is paramount. In contrast, we saw performance gains when an external sound verifier provides the verification signal and critique. We also found that good performance can be achieved without any critique whatsoever: just let the LLM make many guesses, and have a sound verifier pick any that is actually correct.

Our results throw serious doubt on earlier work that was very optimistic about LLM self-critique abilities. They also add depth to contemporaneous studies that focused on benchmarks that were too easy for LLMs to begin with, lacked clear ground truth, and didn't account for test set memorization.

Our proposal, based on the case studies we've performed in this paper is, when possible, to embed LLMs in systems which allow them to guess at solutions multiple times, but which provide some kind of signal for when a guess is good enough. Ideally, this takes the form of a sound verifier, like VAL (Howey et al., 2004) for STRIPS planning, basic expression simplification for Game of 24, or a constraint checker for constraint satisfaction problems. In real-world applications we expect this role to be played by a menagerie of partial critics evaluating plans or solutions based on criteria that they have access to, designed so that consensus is considered verification. Similar architectures have already shown some success (Romera-Paredes et al., 2023), and previous work has proposed the general LLM-Modulo framework (Kambhampati et al., a) which our stance fits into.

## 7 ACKNOWLEDGEMENTS

This research is supported in part by ONR grant N0001423-1-2409, DARPA grant HR00112520016, and gifts from Qualcomm, J.P. Morgan and Amazon.

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

# A   APPENDIX

## A.1   PROMPT VARIATION AND CHAIN OF THOUGHT

LLM results are well-known to be brittle to choice and phrasing of prompt. We ran experiments across multiple prompts to ensure the robustness of our results. Table A1 shows the results for the full pipeline where verification prompts are modified to ask for CoT reasoning first. Results from the main paper are provided for easy comparison. Prompts can be found in each domain's prompt section in this appendix, under the header "Prompt to Elicit CoT Verification".

Performance does improve in some, though not all, cases. However, seemingly near-perfect improvements in verification ability do not translate into near-sound-verifier performance. In G24, CoT increases verification accuracy from 87% to 99%, and does shrink the difference between sound verifier and LLM-verifier in the full pipeline, but a 6 percentage point gap still remains! Furthermore, these improvements come with a large cost increase: in G24, this leads to a 17 times increase in necessary output tokens, which more than doubles the cost of verification.

Note that Chain of Thought techniques themselves vary greatly in their effectiveness across domains. In particular, in reasoning domains like Blocksworld, previous work has shown that they fail to generalize and are not particularly robust (Stechly et al., 2024).

On the rest of the tasks, where verification is also fairly simple and linear, and thus theoretically amenable to CoT, we do not see nearly as significant improvements.

| Domain | S.P. | Sampling | LLM+LLM | LLM+LLM-C | F.E.F. |
|---|---|---|---|---|---|
| Game of 24 | 5% | 28% | 3% | 32% | 38% |
| Coloring | 16% | 40% | 2% | 0% | 37% |
| Blocksworld | 40% | 68% | 55% | 53% | 87% |
| Mystery | 4% | 9% | 0% | 5% | 6% |

Table A1: **Accuracy across prompting schemes including CoT verification schemes over 100 instances per domain.** S.P.–Standard Prompting. Sampling–k=15. LLM+LLM–Main paper result. LLM+LLM-C–Full pipeline with chain of thought verification. F.E.F–(Sound) First Error Feedback.

We also reran our verification-only experiments with these new prompts, as well as with variations on the original (non-CoT) prompts. Table A2 has these results, presented alongside the original ones.

## A.2   ON TREE OF THOUGHTS

Our results on the Game of 24 setting seem to contradict the results shown in (Yao et al., 2023a). However, this is mainly because the self-verification setting presented in the main text of this paper is not directly comparable to that of (Yao et al., 2023a). We ran an additional analysis to provide a direct comparison.

Our external verifier results are all done with only 15 queries to the LLM. (Yao et al., 2023a) isn't entirely clear on the number of queries used, but table 7 in the appendix does give a cost breakdown. Per problem, 100 CoT prompts costs $0.47, but running Tree of Thoughts (ToT) averages $0.74–cost-equivalent to about 150 CoT prompts. On the exact same test set, we extend our experiments to 150 (direct, non-CoT) queries with a sound verifier, and we reach 70%, comparable to ToT's reported 74%.

The remaining difference is likely due to the fact that ToT implements a classical breadth-first search algorithm, only prompting the LLM to generate (much easier) intermediate steps and heuristic evaluations rather than full solutions and reflections. By reducing compositionality and offloading it to a proven classical algorithm, ToT sidesteps some of the major hurdles to LLM reasoning. Our results highlight why such techniques do not scale beyond the simplest toy instances.

| Domain | Accuracy | F.P.R | F.N.R |
|---|---|---|---|
| Game of 24 | 87.0% (3567/4100) | 10.4% (320/3071) | 20.7% (213/1029) |
| Game of 24-CoT | 98.8% (4051/4100) | 0.2% (6/3071) | 4.3% (44/1029) |
| Graph Coloring | 72.4% (362/500) | 6.5% (25/382) | 95.8% (113/118) |
| Graph Coloring-CoT | 77.6% (388/500) | 10.7% (41/382) | 60.2% (71/118) |
| Blocksworld | 71.8% (359/500) | 18.55% (64/345) | 15.48% (24/155) |
| Blocksworld-S | 71.2% (356/500) | 22.1% (76/345) | 8.4% (13/155) |
| Blocksworld-CoT | 77.6% (388/500) | 7.6% (26/345) | 23.9% (37/155) |
| Mystery | 79.6% (398/500) | 0.5% (2/397) | 97.09% (100/103) |
| Mystery-S | 79.0% (395/500) | 1.26% (5/397) | 96.1% (99/103) |
| Mystery-CoT | 81.8% (409/500) | 3.2% (13/397) | 72.8% (75/103) |

Table A2: **LLM Verification results across prompts.** F.P.R. - False Positive Rate, F.N.R - False Negative Rate. The -S cases are non-CoT prompts with the answer and reasoning swapped for those domains where answer was originally asked for first. The -CoT cases are those in which verification is done with CoT.

| | Correct Evaluation | Correct Verification |
|---|---|---|
| correct | 81.6% | 79.1% |
| correct-no-info | 84.4% | - |
| ablated_op | 47.5% | 92.1% |
| ablated_number | 52.2% | 82.9% |
| random | 48.8% | 95.5% |
| random-no-info | 60.3% | - |
| LLM | 55% | 71% |

Table A3: GPT-4's evaluation vs. verification on Game of 24 across expression types.

## A.3 GAME OF 24

### A.3.1 EVALUATION VS. BINARY VERIFICATION FOR GAME OF 24

The following is a more in-depth comparison of GPT-4's critique and verification abilities on game of 24.

For each instance, we generated five different kinds of proposed expressions: correct, ablated operation (exactly one operation is wrong), ablated number (exactly one number is wrong), random, and LLM (sampled from LLM generations). For each of these proposed expression, we sent a query to the LLM asking it to first evaluate the expression and then to say if it is correct, that is equals 24. We also generated two more "no info" cases: correct and random. These two are the exact same as the previous, but only ask for the evaluation of an expression without mentioning the associated goal state (=24) or asking for verification.

Table A.3.1 summarizes the results. Note that we generated 1000 expressions for each type, one from every problem in the full set, but only 100 for the LLM case, as our generations were constrained in the main paper to instances 901-1000.

A.3.2 PROMPTS

All of following examples are built on the same Game of 24 problem, except for the LLM Self-Critique examples.

---

**Raw text format of Game of 24 instance**

```
1 1 4 6
```

---

**Baseline, Direct Prompt**

```
Use numbers and basic arithmetic operations (+ - * /) to obtain
    24. You must write your response. Write your answer first,
    followed by [ANSWER END]
Input: 1 1 4 6
Answer:
```

---

**Example LLM Response**

```
(6 / (1 / 4)) = 24
```

---

**Prompt To Elicit Verification**

```
Please check if the following expression uses only the given
    numbers (and no others) and evaluates to 24:  ((9+10)-(4-9))

Respond only in JSON format as described below:
{
"evaluation": "number the expression evaluated to",
"correct": boolean}
Ensure that Python's json.loads can parse this.Do not provide
    anything else in your response."
```

---

**Prompt To Elicit CoT Verification**

```
Using each of the numbers 1 7 9 11 exactly as many times as they
    appear in the list and the basic arithmetic operations (+ - *
    /), it is possible to write an expression that evaluates to 24.
    Please check if the following expression uses only the correct
    numbers (and no others) and evaluates to 24: (11 - 7) * (9 - 1)
    = 24
If it is not correct, please give feedback on what is wrong and
    how to correct it.
First, think step by step. Check that the expression uses only the
    correct numbers, has exactly the right number of instances each
    number, and evaluates to 24. Then decide what your final answer
    is.
When outputting your final answer, first print the [Answer] tag,
    then put your final answer after the [Answer] tag and respond
    only in JSON format as described below:
{
    "feedback": "feedback",
    "correct": boolean}
Ensure that Python's json.loads can parse this.

Let's think step by step.
[Thoughts]
```

### A.3.3 BACKPROMPTS

---

**Backprompt Generated From Self-Critique**

```
Feedback: This is not correct.
The expression is incorrect. The correct expression should be: (11
    - 2) * 5 + 8 = 24

With this feedback, please try again. Using the numbers 2 5 8 11
    please provide a correct expression that evaluates to 24. Write
    your answer first. At the end of your answer, write [ANSWER END]
Answer: ",
```

---

**Backprompt Generated From Sound Verifier (Pass/Fail)**

```
Feedback: This is not correct. Using the numbers 1 1 4 6 please
    provide a correct expression that evaluates to 24. Write your
    answer first. At the end of your answer, write [ANSWER END]
Answer:
```

---

**Backprompt Generated From Sound Verifier**

```
Feedback: This is not correct. This expression consists of the
    numbers 6, 1, 4, but it has to consist of only and exactly
    ['1', '1', '4', '6']. Using the numbers 1 1 4 6 please provide
    a correct expression that evaluates to 24. Write your answer
    first. At the end of your answer, write [ANSWER END]
Answer: ",
```

---

**Backprompt Generated From Sound Verifier Where the Answer Was Malformed (Missing a Parenthesis)**

```
Use numbers and basic arithmetic operations (+ - * /) to obtain
    24. Write your answer first. At the end of your answer, write
    [ANSWER END]
Input: 1 1 4 6
Answer:
((6 / (1 - 1/4)) = 24[ANSWER END]
Feedback: This is not correct. This expression is malformed. Using
    the numbers 1 1 4 6 please provide a correct expression that
    evaluates to 24. Write your answer first. At the end of your
    answer, write [ANSWER END]
Answer:
```

---

### A.3.4 COMPLETE EXAMPLE OF ITERATIVE PROMPT SEQUENCE

---

**Full Sequence of 7 (Sound Verifier Generated) Backprompts Stopping at Successful Completion of the Task**

```
Use numbers and basic arithmetic operations (+ - * /) to obtain
    24. Write your answer first. At the end of your answer, write
    [ANSWER END]
Input: 1 1 4 6
Answer:
((6 / (1 - 1/4)) = 24[ANSWER END]
Feedback: This is not correct. This expression is malformed. Using
    the numbers 1 1 4 6 please provide a correct expression that
```

---

```
    evaluates to 24. Write your answer first. At the end of your
    answer, write [ANSWER END]
Answer:
((6 / (1 - 1)) * 4 = 24[ANSWER END]
Feedback: This is not correct. This expression is malformed. Using
    the numbers 1 1 4 6 please provide a correct expression that
    evaluates to 24. Write your answer first. At the end of your
    answer, write [ANSWER END]
Answer:
((6 / (1 - 1)) * 4) is undefined because it involves division by
    zero. A correct solution could be:

(6 / (1 / 4)) * 1 = 24[ANSWER END]
Feedback: This is not correct. This expression is malformed. Using
    the numbers 1 1 4 6 please provide a correct expression that
    evaluates to 24. Write your answer first. At the end of your
    answer, write [ANSWER END]
Answer:
(6 / (1 / 4)) = 24[ANSWER END]
Feedback: This is not correct. This expression consists of the
    numbers 6, 1, 4, but it has to consist of only and exactly
    ['1', '1', '4', '6']. Using the numbers 1 1 4 6 please provide
    a correct expression that evaluates to 24. Write your answer
    first. At the end of your answer, write [ANSWER END]
Answer:
(6 - 1) * (4 - 1) = 24[ANSWER END]
Feedback: This is not correct. This expression evaluates to 15
    instead of 24. Using the numbers 1 1 4 6 please provide a
    correct expression that evaluates to 24. Write your answer
    first. At the end of your answer, write [ANSWER END]
Answer:
(6 / (1 / (4 - 1))) = 24[ANSWER END]
Feedback: This is not correct. This expression evaluates to 18
    instead of 24. Using the numbers 1 1 4 6 please provide a
    correct expression that evaluates to 24. Write your answer
    first. At the end of your answer, write [ANSWER END]
Answer:
(6 * (4 - 1)) - 1 = 24[ANSWER END]
Feedback: This is not correct. This expression evaluates to 17
    instead of 24. Using the numbers 1 1 4 6 please provide a
    correct expression that evaluates to 24. Write your answer
    first. At the end of your answer, write [ANSWER END]
Answer:
```

## A.4 GRAPH COLORING

### A.4.1 HALLUCINATIONS IN GRAPH COLORING CRITIQUE

The following is a more in-depth look at GPT-4's critique abilities on the graph coloring problem.

For each instance, we generated five different kinds of colorings: correct, ablated (exactly one edge is wrong), non-optimal (a coloring with no constraint violations except that it uses too many colors), random, and LLM (sampled from LLM generations). For each of these 500 proposed colorings, we sent a query to the LLM asking it to verify if the answer was right, and–if not–to output a free-form critique. We then parsed these to determine what edges it said were colored the same at both vertices.

Table A4 summarizes the results. Note that, proportionally, hallucinations decrease when the error rate of the domain increases. That is to say, when there are more incorrect edges, the model is more likely to point to one of them. Intuitively, this makes sense: it's easier to guess one edge which is wrong when half of all the edges are miscolored, as is the case on average among randomly colored graphs.

Table A4: Distribution of hallucinations during verification task. This table counts the number of instances that featured each type of hallucination and compares it to the total number of erroneous edges encountered across all coloring instances in each subset. Note that the correct column tracks the number of colorings where the *ground truth* is that the coloring is correct.

| | Hallucinations | | | | Coloring | |
|---|---|---|---|---|---|---|
| | Vertex | Edge | Both | None | Errors | Correct |
| Correct | 29 | 72 | 7 | 2 | 0 | 100 |
| Ablated | 24 | 52 | 5 | 24 | 187 | 0 |
| Non-optimal | 18 | 65 | 3 | 10 | 0 | 0 |
| Random | 10 | 26 | 5 | 66 | 736 | 0 |
| LLM | 26 | 41 | 6 | 27 | 240 | 18 |
| Total | 107 | 256 | 26 | 129 | 282 | 118 |

Edge hallucinations are more common than vertex. Essentially, typical behavior is to pick two vertices that are the same color in the coloring, but which aren't associated by an edge in the graph description, and claim that they are connected and thus illegally colored. Vertex color hallucination is when the reverse happens: instead of ascribing an edge to same-color nodes, the colorings of two connected vertices are misstated. The overlap between the two cases, where a non-existent edge is declared to be violated by non-existent colorings is much rarer than either. Note that it never hallucinates new vertex *names*, only that vertices which are in graph have colors differing from reality.

Even rarer cases did spring up in the response data. At times the model lost track of the question being asked and reversed it, explicitly claiming that two same-colored vertices violate the conditions because they *aren't* connected; or it began to contradict itself mid-deduction, making multiple claims about a vertex's color.

We present these examples here. In the following, multiple equals signs in a row ("===") are dividers between separate examples, not parts of the examples.

---

**Examples of LLM Output on the Verification Task**

```
The vertices 1 and 4 are both color 0 and share an edge.
============================
The vertices 9 and 12 are both colored with Color 1 and they share
    an edge.
============================
Vertex 1: Color 2 is connected to Vertex 2: Color 2.
============================
Vertex 5 and vertex 10 are both Color 2 and share an edge.
============================
Same-color vertices 0 and 1 share an edge.
============================
Vertex 0 (Color 0) and vertex 3 (Color 0) share an edge. Vertex 1
    (Color 1) and vertex 2 (Color 1) share an edge. Vertex 4 (Color
    1) and vertex 8 (Color 1) share an edge. Vertex 2 (Color 1) and
    vertex 4 (Color 1) share an edge. Vertex 4 (Color 1) and vertex
    11 (Color 0) share an edge. Vertex 12 (Color 2) and vertex 15
    (Color 2) share an edge.
The vertices 3 and 0 both have Color 2 and they are adjacent. The
    vertices 1 and 11 both have Color 1 and they are adjacent. The
    vertices 2 and 5 both have Color 0 and they are adjacent.
    Therefore, the given coloring is incorrect.
============================
0: Color 0 and 9: Color 0 share an edge.
0: Color 0 and 10: Color 0 share an edge.
0: Color 0 and 14: Color 0 share an edge.
9: Color 0 and 13: Color 0 share an edge.
============================
```

```
Vertex 3 is colored with Color 4, which exceeds the optimal
    coloring number of 4. Therefore, this coloring is non-optimal.
==============================
Vertex 0 is missing in the coloring. Missing assignment.
==============================
Vertex 0 (color 3) is connected to vertex 3 (color 2), sharing an
    edge. Vertex 0 (color 3) is also connected to vertex 4 (color
    0), and to vertex 5 (color 0), and to vertex 9 (color 0), and
    to vertex 12 (color 2), sharing an edge with them. Therefore,
    this assignment is correctly colored with color 0, color 2, and
    color 3, making use of 3 different colors.

Vertex 1 (color 2) is connected to vertex 4 (color 0), and to
    vertex 5 (color 0), and to vertex 6 (color 0), and to vertex 7
    (color 3), and to vertex 8 (color 0), and to vertex 10 (color
    1), and to vertex 11 (color 1), sharing an edge with them.
    Therefore, this assignment is also correctly colored.

Vertex 2 (color 0) is connected to vertex 12 (color 2), sharing an
    edge. Therefore, this assignment is correctly colored.

Vertex 3 (color 2) is connected to vertex 5 (color 0), and to
    vertex 8 (color 0), and to vertex 10 (color 1), sharing an edge
    with them. Therefore, this assignment is correctly colored.

Vertex 4 (color 0) is connected to vertex 7 (color 3), and to
    vertex 11 (color 1), sharing an edge. Therefore, this
    assignment is correctly colored.

Vertex 5 (color 0) is connected to vertex 7 (color 3), sharing an
    edge. Therefore, this assignment is correctly colored.

Vertex 6 (color 0) is connected to vertex 10 (color 1), and to
    vertex 12 (color 2), sharing an edge. Therefore, this
    assignment is correctly colored.

Vertex 8 (color 0) is connected to vertex 10 (color 1), sharing an
    edge. Therefore, this assignment is correctly colored.

Vertex 9 (color 0) is connected to vertex 11 (color 1), sharing an
    edge. Therefore, this assignment is correctly colored.

Vertex 10 (color 1) is connected to vertex 12 (color 2), sharing
    an edge. Therefore, this assignment is correctly colored.

Vertex 11 (color 1) isn't connected to vertex 10 (color 1), hence,
    this assignment is correctly colored.

Vertex 12 (color 2) isn't connected to vertex 2 (color 0), hence,
    this assignment is correctly colored.

No adjacent vertices share a same color, and no colors exceed the
    optimal coloring number. Hence, the coloring is correct.
Verifier confirmed success.
```

**Examples of (Rare) Mangled, Illogical, and Otherwise Strange Output on the Verification Task**

```
Vertex 10, colored 4, is connected to vertex 11, also colored 1.
    However, it is also colored 4, which is greater than the
    optimal coloring of 4. So, this coloring is non-optimal.
==============================
```

```
Vertex 9 (Color 2) and Vertex 10 (Color 1) are connected and share
    the same color. Vertex 9 (Color 2) and Vertex 11 (Color 1) are
    also connected and share the same color.
============================
Vertex 2 (Color 0) is connected to Vertex 0 (Color 0).
Vertex 5 (Color 0) is connected to Vertex 1 (Color 2) and Vertex
    14 (Color 1), but also to Vertex 7 (Color 1).
Vertex 6 (Color 0) is connected to Vertex 3 (Color 2), Vertex 10
    (Color 1) and 13 (Color 2), but also to Vertex 8 (Color 2).
============================
The coloring is incorrect. Vertex 1 (Color 2) and Vertex 3 (Color
    3) are connected but have different colors. Vertex 3 (Color 3)
    is connected to Vertex 6 (Color 3).
============================
Vertex 10, colored 4, is connected to vertex 11, also colored 1.
    However, it is also colored 4, which is greater than the
    optimal coloring of 4. So, this coloring is non-optimal.
```

### A.4.2  PROMPTS

All of following examples are built on the same graph instance.

#### DIMACS Format For Graphs

```
e 0 7
e 0 8
e 0 9
e 0 11
e 1 13
e 2 9
e 3 8
e 3 11
e 3 12
e 4 12
e 5 11
e 6 9
e 7 10
e 7 13
e 9 11
e 10 13
e 11 13
c OPTIMAL CHROMATIC NUMBER === 3
```

#### Baseline, Direct Prompt

```
Color the following graph, described as a set of edges, such that
    no two vertices on the same edge share a color.
You may use at most 3 colors.
Vertex 0 is connected to vertex 7.
Vertex 0 is connected to vertex 8.
Vertex 0 is connected to vertex 9.
Vertex 0 is connected to vertex 11.
Vertex 1 is connected to vertex 13.
Vertex 2 is connected to vertex 9.
Vertex 3 is connected to vertex 8.
Vertex 3 is connected to vertex 11.
Vertex 3 is connected to vertex 12.
Vertex 4 is connected to vertex 12.
Vertex 5 is connected to vertex 11.
```

```
Vertex 6 is connected to vertex 9.
Vertex 7 is connected to vertex 10.
Vertex 7 is connected to vertex 13.
Vertex 9 is connected to vertex 11.
Vertex 10 is connected to vertex 13.
Vertex 11 is connected to vertex 13.
There are a total of 14 vertices. Please label every vertex, even
    if it is disconnected from the rest of the graph.Please provide
    each vertex's color. Do not skip any vertices. Each color must
    be provided on a new line in the response and should be
    formatted as "{VERTEX NUMBER}: {VERTEX COLOR ASSIGNMENT}".
    Please do not provide anything else in your response.
```

## Example LLM Response

```
0: Color1
1: Color1
2: Color1
3: Color2
4: Color1
5: Color2
6: Color2
7: Color2
8: Color3
9: Color3
10: Color3
11: Color3
12: Color3
13: Color1
```

## Prompt To Elicit Verification

```
The following graph, described as a set of edges, has an optimal
    coloring number of 3:
Vertex 0 is connected to vertex 7.
Vertex 0 is connected to vertex 8.
Vertex 0 is connected to vertex 9.
Vertex 0 is connected to vertex 11.
Vertex 1 is connected to vertex 13.
Vertex 2 is connected to vertex 9.
Vertex 3 is connected to vertex 8.
Vertex 3 is connected to vertex 11.
Vertex 3 is connected to vertex 12.
Vertex 4 is connected to vertex 12.
Vertex 5 is connected to vertex 11.
Vertex 6 is connected to vertex 9.
Vertex 7 is connected to vertex 10.
Vertex 7 is connected to vertex 13.
Vertex 9 is connected to vertex 11.
Vertex 10 is connected to vertex 13.
Vertex 11 is connected to vertex 13.
Please check if this coloring is correct:0: Color1
1: Color1
2: Color1
3: Color2
4: Color1
5: Color2
6: Color2
7: Color2
```

```
8: Color3
9: Color3
10: Color3
11: Color3
12: Color3
13: Color1
If it is, say 'Verifier confirmed success.' Do not provide
    anything else in your response. If it is incorrect, please
    point out which same-color vertices share an edge.
```

**Prompt To Elicit CoT Verification**

```
[Instructions]
When outputting your final answer, first print the [Answer] tag,
    then put your final answer after the [Answer] tag. Respond only
    in the following format:
Wrong Edges: a list of incorrect edges
All Vertices Colored: boolean representing if every vertex is
    colored
Optimal Or Less: boolean representing if the number of colors is
    no more than the optimal
Correct: boolean

[Graph]
The following graph, described as a set of edges, has an optimal
    coloring number of 3:
Vertex 0 is connected to vertex 7.
Vertex 0 is connected to vertex 8.
Vertex 0 is connected to vertex 9.
Vertex 0 is connected to vertex 11.
Vertex 1 is connected to vertex 13.
Vertex 2 is connected to vertex 9.
Vertex 3 is connected to vertex 8.
Vertex 3 is connected to vertex 11.
Vertex 3 is connected to vertex 12.
Vertex 4 is connected to vertex 12.
Vertex 5 is connected to vertex 11.
Vertex 6 is connected to vertex 9.
Vertex 7 is connected to vertex 10.
Vertex 7 is connected to vertex 13.
Vertex 9 is connected to vertex 11.
Vertex 10 is connected to vertex 13.
Vertex 11 is connected to vertex 13.

[Coloring]
A coloring is correct if no adjacent vertices are the same color
    and the total number of colors used is no more than the optimal
    coloring number. Please check if this coloring is correct: 0:
    Color1
1: Color1
2: Color1
3: Color1
4: Color1
5: Color1
6: Color1
7: Color2
8: Color2
9: Color2
10: Color1
11: Color3
12: Color2
```

```
13: Color3
[ANSWER END]

Let's think step by step. Remember to output your final answer in
    the format described in the instructions.
[Thoughts]
```

### A.4.3 BACKPROMPTS

**Backprompt Generated From Self-Critique**

```
This is incorrect. Feedback:
Vertices 0 and 3 share an edge and are both colored with Color1.
    Vertices 9 and 11 share an edge and are both colored with
    Color3.

Using this feedback, please try again. Please provide each
    vertex's color. Do not skip any vertices. Each color must be
    provided on a new line in the response and should be formatted
    as "{VERTEX NUMBER}: {VERTEX COLOR ASSIGNMENT}". Please do not
    provide anything else in your response.
```

**Backprompt Generated From Sound Verifier (Pass/Fail)**

```
This is not correct. Using the previously provided graph, please
    provide a correct coloring. Please provide each vertex's color.
    Do not skip any vertices. Each color must be provided on a new
    line in the response and should be formatted as "{VERTEX
    NUMBER}: {VERTEX COLOR ASSIGNMENT}". Please do not provide
    anything else in your response.
```

**Backprompt Generated From Sound Verifier (First)**

```
Vertex 9 and vertex 11 were both colored Color3 despite being
    connected by an edge.
This is wrong. Please recolor. Please provide each vertex's color.
    Do not skip any vertices. Each color must be provided on a new
    line in the response and should be formatted as "{VERTEX
    NUMBER}: {VERTEX COLOR ASSIGNMENT}". Please do not provide
    anything else in your response.
```

**Backprompt Generated From Sound Verifier (Full)**

```
Vertex 9 and vertex 11 were both colored Color3 despite being
    connected by an edge. Vertex 10 and vertex 13 were both colored
    Color3 despite being connected by an edge. Vertex 11 and vertex
    13 were both colored Color3 despite being connected by an edge.
This is wrong. Please recolor. Please provide each vertex's color.
    Do not skip any vertices. Each color must be provided on a new
    line in the response and should be formatted as "{VERTEX
    NUMBER}: {VERTEX COLOR ASSIGNMENT}". Please do not provide
    anything else in your response.
```

A.4.4 COMPLETE EXAMPLE OF ITERATIVE PROMPT SEQUENCE

---

**Full Sequence of 15 (LLM-Generated) Backprompts**

```
Color the following graph, described as a set of edges, such that
    no two vertices on the same edge share a color.
You may use at most 3 colors.
Vertex 0 is connected to vertex 7.
Vertex 0 is connected to vertex 8.
Vertex 0 is connected to vertex 9.
Vertex 0 is connected to vertex 11.
Vertex 1 is connected to vertex 13.
Vertex 2 is connected to vertex 9.
Vertex 3 is connected to vertex 8.
Vertex 3 is connected to vertex 11.
Vertex 3 is connected to vertex 12.
Vertex 4 is connected to vertex 12.
Vertex 5 is connected to vertex 11.
Vertex 6 is connected to vertex 9.
Vertex 7 is connected to vertex 10.
Vertex 7 is connected to vertex 13.
Vertex 9 is connected to vertex 11.
Vertex 10 is connected to vertex 13.
Vertex 11 is connected to vertex 13.
There are a total of 14 vertices. Please label every vertex, even
    if it is disconnected from the rest of the graph.Please provide
    each vertex's color. Do not skip any vertices. Each color must
    be provided on a new line in the response and should be
    formatted as "{VERTEX NUMBER}: {VERTEX COLOR ASSIGNMENT}".
    Please do not provide anything else in your response.

0: Color1
1: Color1
2: Color1
3: Color1
4: Color1
5: Color2
6: Color2
7: Color2
8: Color2
9: Color3
10: Color3
11: Color3
12: Color2
13: Color3

This is incorrect. Feedback:
Vertices 0 and 3 share an edge and are both colored with Color1.
    Vertices 9 and 11 share an edge and are both colored with
    Color3.

Using this feedback, please try again. Please provide each
    vertex's color. Do not skip any vertices. Each color must be
    provided on a new line in the response and should be formatted
    as "{VERTEX NUMBER}: {VERTEX COLOR ASSIGNMENT}". Please do not
    provide anything else in your response.

0: Color1
1: Color1
2: Color2
3: Color2
4: Color2
5: Color1
```

```
6: Color1
7: Color2
8: Color3
9: Color3
10: Color1
11: Color2
12: Color3
13: Color3

This is incorrect. Feedback:
Vertices 2 and 3 are both colored with Color2 and share an edge.
    Vertices 7 and 11 are both colored with Color2 and share an
    edge. Vertices 9 and 13 are both colored with Color3 and share
    an edge.

Using this feedback, please try again. Please provide each
    vertex's color. Do not skip any vertices. Each color must be
    provided on a new line in the response and should be formatted
    as "{VERTEX NUMBER}: {VERTEX COLOR ASSIGNMENT}". Please do not
    provide anything else in your response.

0: Color1
1: Color1
2: Color2
3: Color3
4: Color3
5: Color3
6: Color3
7: Color3
8: Color2
9: Color1
10: Color2
11: Color2
12: Color1
13: Color1

This is incorrect. Feedback:
Vertices 0 and 9 share an edge and are both colored with Color1.
    Vertices 0 and 11 share an edge and are both colored with
    Color2. Vertices 1 and 13 share an edge and are both colored
    with Color1. Vertices 7 and 13 share an edge and are both
    colored with Color3.

Using this feedback, please try again. Please provide each
    vertex's color. Do not skip any vertices. Each color must be
    provided on a new line in the response and should be formatted
    as "{VERTEX NUMBER}: {VERTEX COLOR ASSIGNMENT}". Please do not
    provide anything else in your response.

0: Color1
1: Color1
2: Color2
3: Color2
4: Color3
5: Color1
6: Color1
7: Color2
8: Color3
9: Color3
10: Color1
11: Color3
12: Color1
13: Color3
```

```
This is incorrect. Feedback:
Vertices 0 and 5 both have Color1 and share an edge. Vertices 0
    and 11 both have Color3 and share an edge. Vertices 7 and 10
    both have Color2 and share an edge. Vertices 11 and 13 both
    have Color3 and share an edge.

Using this feedback, please try again. Please provide each
    vertex's color. Do not skip any vertices. Each color must be
    provided on a new line in the response and should be formatted
    as "{VERTEX NUMBER}: {VERTEX COLOR ASSIGNMENT}". Please do not
    provide anything else in your response.

0: Color1
1: Color1
2: Color2
3: Color2
4: Color1
5: Color1
6: Color1
7: Color2
8: Color3
9: Color3
10: Color3
11: Color2
12: Color3
13: Color1

This is incorrect. Feedback:
Vertices 0 and 1 both have Color1 and share an edge. Vertices 0
    and 5 also both have Color1 and share an edge. Vertices 0 and 6
    also both have Color1 and share an edge. Vertices 7 and 11 both
    have Color2 and share an edge. Vertices 9 and 11 both have
    Color2 and share an edge.

Using this feedback, please try again. Please provide each
    vertex's color. Do not skip any vertices. Each color must be
    provided on a new line in the response and should be formatted
    as "{VERTEX NUMBER}: {VERTEX COLOR ASSIGNMENT}". Please do not
    provide anything else in your response.

0: Color1
1: Color2
2: Color2
3: Color2
4: Color2
5: Color3
6: Color3
7: Color3
8: Color2
9: Color3
10: Color1
11: Color2
12: Color3
13: Color1

This is incorrect. Feedback:
Vertices 0 and 8 both have Color1 and share an edge. Vertices 0
    and 11 both have Color1 and share an edge. Vertices 3 and 8
    both have Color2 and share an edge. Vertices 3 and 11 both have
    Color2 and share an edge. Vertices 7 and 13 both have Color3
    and share an edge. Vertices 9 and 11 both have Color3 and share
    an edge. Vertices 11 and 13 both have Color2 and share an edge.
```

```
Using this feedback, please try again. Please provide each
    vertex's color. Do not skip any vertices. Each color must be
    provided on a new line in the response and should be formatted
    as "{VERTEX NUMBER}: {VERTEX COLOR ASSIGNMENT}". Please do not
    provide anything else in your response.

0: Color1
1: Color2
2: Color2
3: Color3
4: Color1
5: Color1
6: Color1
7: Color3
8: Color2
9: Color3
10: Color2
11: Color2
12: Color3
13: Color1

This is incorrect. Feedback:
Vertices 0 and 11 share an edge and are both colored with Color1.
    Vertices 2 and 9 share an edge and are both colored with
    Color2. Vertices 3 and 12 share an edge and are both colored
    with Color3.

Using this feedback, please try again. Please provide each
    vertex's color. Do not skip any vertices. Each color must be
    provided on a new line in the response and should be formatted
    as "{VERTEX NUMBER}: {VERTEX COLOR ASSIGNMENT}". Please do not
    provide anything else in your response.

0: Color1
1: Color2
2: Color2
3: Color2
4: Color3
5: Color3
6: Color3
7: Color2
8: Color3
9: Color1
10: Color3
11: Color3
12: Color1
13: Color1

This is incorrect. Feedback:
Vertices 11 and 13 both have Color3 and share an edge.

Using this feedback, please try again. Please provide each
    vertex's color. Do not skip any vertices. Each color must be
    provided on a new line in the response and should be formatted
    as "{VERTEX NUMBER}: {VERTEX COLOR ASSIGNMENT}". Please do not
    provide anything else in your response.

0: Color1
1: Color2
2: Color2
3: Color2
4: Color3
```

```
5: Color3
6: Color3
7: Color2
8: Color3
9: Color1
10: Color3
11: Color1
12: Color1
13: Color2

This is incorrect. Feedback:
Vertices 0 and 11 share an edge and both are colored with Color1.

Using this feedback, please try again. Please provide each
    vertex's color. Do not skip any vertices. Each color must be
    provided on a new line in the response and should be formatted
    as "{VERTEX NUMBER}: {VERTEX COLOR ASSIGNMENT}". Please do not
    provide anything else in your response.

0: Color1
1: Color2
2: Color2
3: Color3
4: Color1
5: Color2
6: Color3
7: Color3
8: Color2
9: Color3
10: Color1
11: Color2
12: Color2
13: Color1

This is incorrect. Feedback:
Vertices 2 and 9 share an edge and are both colored with Color2.
    Similarly, vertices 11 and 13 share an edge and are both
    colored with Color2.

Using this feedback, please try again. Please provide each
    vertex's color. Do not skip any vertices. Each color must be
    provided on a new line in the response and should be formatted
    as "{VERTEX NUMBER}: {VERTEX COLOR ASSIGNMENT}". Please do not
    provide anything else in your response.

0: Color1
1: Color2
2: Color2
3: Color3
4: Color1
5: Color2
6: Color3
7: Color3
8: Color2
9: Color1
10: Color2
11: Color3
12: Color2
13: Color1

This is incorrect. Feedback:
Vertices 2 and 9 share an edge and are both colored with Color2.
    Vertices 3 and 11 share an edge and are both colored with
```

```
    Color3. Vertices 7 and 13 share an edge and are both colored
    with Color3.

Using this feedback, please try again. Please provide each
    vertex's color. Do not skip any vertices. Each color must be
    provided on a new line in the response and should be formatted
    as "{VERTEX NUMBER}: {VERTEX COLOR ASSIGNMENT}". Please do not
    provide anything else in your response.

0: Color1
1: Color2
2: Color2
3: Color3
4: Color1
5: Color2
6: Color3
7: Color2
8: Color3
9: Color3
10: Color1
11: Color2
12: Color3
13: Color1

This is incorrect. Feedback:
Vertices 0 and 11 share an edge and are both colored with Color1.
    Vertices 3 and 8 share an edge and are both colored with
    Color3. Vertices 7 and 13 share an edge and are both colored
    with Color2.

Using this feedback, please try again. Please provide each
    vertex's color. Do not skip any vertices. Each color must be
    provided on a new line in the response and should be formatted
    as "{VERTEX NUMBER}: {VERTEX COLOR ASSIGNMENT}". Please do not
    provide anything else in your response.

0: Color1
1: Color2
2: Color2
3: Color3
4: Color1
5: Color2
6: Color3
7: Color2
8: Color3
9: Color1
10: Color3
11: Color2
12: Color1
13: Color3

This is incorrect. Feedback:
The coloring is incorrect. Vertices 0 and 11 share an edge and are
    both colored with Color1. Vertices 7 and 13 share an edge and
    are both colored with Color2.

Using this feedback, please try again. Please provide each
    vertex's color. Do not skip any vertices. Each color must be
    provided on a new line in the response and should be formatted
    as "{VERTEX NUMBER}: {VERTEX COLOR ASSIGNMENT}". Please do not
    provide anything else in your response.

0: Color1
```

```
1: Color2
2: Color2
3: Color3
4: Color1
5: Color2
6: Color3
7: Color3
8: Color2
9: Color1
10: Color2
11: Color3
12: Color1
13: Color1

This is incorrect. Feedback:
Vertices 0 and 9 share an edge and are both colored with Color1.
    Vertices 7 and 11 share an edge and are both colored with
    Color3. Vertices 1 and 13 share an edge and are both colored
    with Color2.

Using this feedback, please try again. Please provide each
    vertex's color. Do not skip any vertices. Each color must be
    provided on a new line in the response and should be formatted
    as "{VERTEX NUMBER}: {VERTEX COLOR ASSIGNMENT}". Please do not
    provide anything else in your response.

0: Color1
1: Color2
2: Color3
3: Color2
4: Color2
5: Color1
6: Color2
7: Color3
8: Color3
9: Color2
10: Color1
11: Color3
12: Color1
13: Color1

This is incorrect. Feedback:
Vertices 3 and 9 share an edge and are both colored with Color2.

Using this feedback, please try again. Please provide each
    vertex's color. Do not skip any vertices. Each color must be
    provided on a new line in the response and should be formatted
    as "{VERTEX NUMBER}: {VERTEX COLOR ASSIGNMENT}". Please do not
    provide anything else in your response.

0: Color1
1: Color2
2: Color3
3: Color2
4: Color3
5: Color1
6: Color2
7: Color3
8: Color2
9: Color1
10: Color2
11: Color3
12: Color1
```

| | Binary Verification | Type Identified | Critique Generation |
|---|---|---|---|
| Correct | 78/100 (78%) | 78/100 (78%) | 78/100 (78%) |
| Inexecutable | 70/100 (70%) | 63/100 (63%) | 8/100 (8%) |
| Non Goal Reaching | 98/100 (98%) | 12/100 (12%) | 12/100 (12%) |
| Random | 100/100 (100%) | 94/100 (94%) | 2/100 (2%) |
| LLM Correct | 53/55 (96.36%) | 53/55 (96.36%) | 53/55 (96.36%) |
| LLM Inexecutable | 25/40 (62.5%) | 24/40 (60%) | 0/40 (0%) |
| LLM Non Goal Reaching | 3/5 (60%) | 2/5 (40%) | 2/5 (40%) |

Table A5: This table presents the verification and critique accuracy of LLM-as-verifier across five different kinds of plans over 100 instances of Blocksworld.

| | Binary Verification | Type Identified | Critique Generation |
|---|---|---|---|
| Correct | 3/100 (3%) | 3/100 (3%) | 3/100 (3%) |
| Inexecutable | 100/100 (100%) | 100/100 (100%) | 24/100 (24%) |
| Non Goal Reaching | 98/100 (98%) | 12/100 (12%) | 12/100 (12%) |
| Random | 100/100 (100%) | 100/100 (100%) | 59/100 (100%) |
| LLM Correct | 0/3 (0%) | 0/3 (0%) | 0/3 (0%) |
| LLM Inexecutable | 89/89 (100%) | 89/89 (100%) | 12/89 (13.48%) |
| LLM Non Goal Reaching | 8/8 (100%) | 0/8 (0%) | 0/8 (0%) |

Table A6: This table presents the verification and critique accuracy of LLM-as-verifier across five different kinds of plans over 100 instances of Mystery Blocksworld.

```
13: Color1
```

## A.5    PLANNING

### A.5.1    ACCURACY OF PLANNING CRITIQUE

For each instance, we generated five different kind of plans: correct, inexecutable (where an action is inexecutable and the plan is thus invalid), non goal reaching (all actions are executable but the plan does not reach the desired state), random, and LLM (sampled from the LLM generations). For each of these prompts, we sent a query to the LLM asking it to verify the plan and critique it in a certain format. We provide an example in the following prompts section. Specifically, if the plan is valid, the LLM has to just output that. If a plan is invalid and inexecutable, it has to provide the first action that the plan is rendered invalid and the unmet preconditions for that action. If the plan is invalid and non goal reaching, it has to provide the unmet goals for the plan. We evaluate the binary verification and the critique by comparing it to the ground truth provided by VAL (Howey et al., 2004). We check the binary verification, whether or not the detected the type of invalidity (inexecutable or non goal reaching) and if the critique is valid. Tables A5 and A6 show the inability of LLMs in generating the right critique for proposed solutions even though they get the label correct.

### A.5.2    PROMPTS - BLOCKSWORLD

All of following examples are built on the same graph instance.

---

**Baseline, Direct Prompt**

```
I am playing with a set of blocks where I need to arrange the
    blocks into stacks. Here are the actions I can do

Pick up a block
Unstack a block from on top of another block
Put down a block
Stack a block on top of another block

I have the following restrictions on my actions:
I can only pick up or unstack one block at a time.
I can only pick up or unstack a block if my hand is empty.
I can only pick up a block if the block is on the table and the
    block is clear. A block is clear if the block has no other
    blocks on top of it and if the block is not picked up.
I can only unstack a block from on top of another block if the
    block I am unstacking was really on top of the other block.
I can only unstack a block from on top of another block if the
    block I am unstacking is clear.
Once I pick up or unstack a block, I am holding the block.
I can only put down a block that I am holding.
I can only stack a block on top of another block if I am holding
    the block being stacked.
I can only stack a block on top of another block if the block onto
    which I am stacking the block is clear.
Once I put down or stack a block, my hand becomes empty.
Once you stack a block on top of a second block, the second block
    is no longer clear.

[STATEMENT]
As initial conditions I have that, the red block is clear, the
    yellow block is clear, the hand is empty, the red block is on
    top of the blue block, the yellow block is on top of the orange
    block, the blue block is on the table and the orange block is
    on the table.
My goal is to have that the orange block is on top of the red
    block.

My plan is as follows:

[PLAN]
unstack the yellow block from on top of the orange block
put down the yellow block
pick up the orange block
stack the orange block on top of the red block
[PLAN END]

[STATEMENT]
As initial conditions I have that, the blue block is clear, the
    hand is empty, the blue block is on top of the orange block,
    the orange block is on top of the yellow block, the yellow
    block is on top of the red block and the red block is on the
    table.
My goal is to have that the red block is on top of the orange
    block and the yellow block is on top of the red block.

My plan is as follows:

[PLAN]
```

Example LLM Response

```
unstack the blue block from on top of the orange block
put down the blue block
unstack the orange block from on top of the yellow block
put down the orange block
unstack the yellow block from on top of the red block
put down the yellow block
pick up the red block
stack the red block on top of the orange block
pick up the yellow block
stack the yellow block on top of the red block
[PLAN END]
```

Prompt To Elicit Verification (Open Ended)

```
I am playing with a set of blocks where I need to arrange the
    blocks into stacks. Here are the actions I can do

Pick up a block
Unstack a block from on top of another block
Put down a block
Stack a block on top of another block

I have the following restrictions on my actions:
I can only pick up or unstack one block at a time.
I can only pick up or unstack a block if my hand is empty.
I can only pick up a block if the block is on the table and the
    block is clear. A block is clear if the block has no other
    blocks on top of it and if the block is not picked up.
I can only unstack a block from on top of another block if the
    block I am unstacking was really on top of the other block.
I can only unstack a block from on top of another block if the
    block I am unstacking is clear.
Once I pick up or unstack a block, I am holding the block.
I can only put down a block that I am holding.
I can only stack a block on top of another block if I am holding
    the block being stacked.
I can only stack a block on top of another block if the block onto
    which I am stacking the block is clear.
Once I put down or stack a block, my hand becomes empty.
Once you stack a block on top of a second block, the second block
    is no longer clear.

[STATEMENT]
As initial conditions I have that, the blue block is clear, the
    orange block is clear, the hand is empty, the orange block is
    on top of the yellow block, the yellow block is on top of the
    red block, the red block is on the table and the blue block is
    on the table.
My goal is to have that the red block is on top of the orange
    block, the blue block is on top of the red block and the yellow
    block is on top of the blue block.
My plan is as follows:

[PLAN]
unstack the orange block from on top of the yellow block
put down the orange block
unstack the yellow block from on top of the red block
put down the yellow block
pick up the blue block
stack the blue block on top of the red block
```

```
pick up the orange block
stack the orange block on top of the blue block
pick up the yellow block
stack the yellow block on top of the orange block
[PLAN END]

Verify whether the above plan is valid. If it is valid, please say
    "Plan is valid." and nothing else. If it is invalid, please say
    "Plan is invalid." and then provide feedback on why the plan
    fails.
```

## Prompt To Elicit Verification (Format based)

```
I am playing with a set of blocks where I need to arrange the
    blocks into stacks. Here are the actions I can do

Pick up a block
Unstack a block from on top of another block
Put down a block
Stack a block on top of another block

I have the following restrictions on my actions:
I can only pick up or unstack one block at a time.
I can only pick up or unstack a block if my hand is empty.
I can only pick up a block if the block is on the table and the
    block is clear. A block is clear if the block has no other
    blocks on top of it and if the block is not picked up.
I can only unstack a block from on top of another block if the
    block I am unstacking was really on top of the other block.
I can only unstack a block from on top of another block if the
    block I am unstacking is clear.
Once I pick up or unstack a block, I am holding the block.
I can only put down a block that I am holding.
I can only stack a block on top of another block if I am holding
    the block being stacked.
I can only stack a block on top of another block if the block onto
    which I am stacking the block is clear.
Once I put down or stack a block, my hand becomes empty.
Once you stack a block on top of a second block, the second block
    is no longer clear.

[STATEMENT]
As initial conditions I have that, the blue block is clear, the
    orange block is clear, the hand is empty, the orange block is
    on top of the yellow block, the yellow block is on top of the
    red block, the red block is on the table and the blue block is
    on the table.
My goal is to have that the red block is on top of the orange
    block, the blue block is on top of the red block and the yellow
    block is on top of the blue block.
My plan is as follows:

[PLAN]
unstack the orange block from on top of the yellow block
put down the orange block
unstack the yellow block from on top of the red block
put down the yellow block
pick up the red block
stack the red block on top of the orange block
pick up the blue block
stack the blue block on top of the red block
```

```
pick up the yellow block
stack the yellow block on top of the blue block
[PLAN END]

Verify whether the above plan is valid. Provide a JSON between
    tags [JSON] and [JSON_END] for the verification information.
    The JSON should contain three main keys: (1) "valid": a binary
    value that tells if the plan is valid or not i.e., the plan
    when executed satisfies the goal conditions. If the plan is
    invalid and inexecutable then include (2)
    "unmet_preconditions": This contains two more keys; (2.1)
    "action": This is the name of the first action that renders the
    plan inexecutable (2.2) "preconditions": A list of unmet
    preconditions for the mentioned action; If the plan is
    executable but not goal reaching then include (3)
    "unmet_goals": A list of unmet goal conditions in the JSON.
    Include only one of the keys (2) or (3) based on the plan
    invalidity.
```

### Prompt To Elicit Verification (Chain of thought based)

```
I am playing with a set of blocks where I need to arrange the
    blocks into stacks. Here are the actions I can do

Pick up a block
Unstack a block from on top of another block
Put down a block
Stack a block on top of another block

I have the following restrictions on my actions:
I can only pick up or unstack one block at a time.
I can only pick up or unstack a block if my hand is empty.
I can only pick up a block if the block is on the table and the
    block is clear. A block is clear if the block has no other
    blocks on top of it and if the block is not picked up.
I can only unstack a block from on top of another block if the
    block I am unstacking was really on top of the other block.
I can only unstack a block from on top of another block if the
    block I am unstacking is clear.
Once I pick up or unstack a block, I am holding the block.
I can only put down a block that I am holding.
I can only stack a block on top of another block if I am holding
    the block being stacked.
I can only stack a block on top of another block if the block onto
    which I am stacking the block is clear.
Once I put down or stack a block, my hand becomes empty.
Once you stack a block on top of a second block, the second block
    is no longer clear.

[STATEMENT]
As initial conditions I have that, the red block is clear, the
    yellow block is clear, the hand is empty, the red block is on
    top of the blue block, the yellow block is on top of the orange
    block, the blue block is on the table and the orange block is
    on the table.
My goal is to have that the orange block is on top of the red
    block.
My plan is as follows:

[PLAN]
unstack the yellow block from on top of the orange block
```

```
put down the yellow block
pick up the orange block
stack the orange block on top of the red block
[PLAN END]

Verify whether the above plan is valid. You will think step by
    step and output intermediate reasoning steps and thoughts for
    the verification after the [THOUGHTS] tag. Then, provide a JSON
    between the tags [JSON] and [JSON_END] for the verification
    information. The JSON should contain three main keys: If the
    plan is invalid and inexecutable then include (1)
    "unmet_preconditions": This contains two more keys; (1.1)
    "action": This is the name of the first action that renders the
    plan inexecutable (1.2) "preconditions": A list of unmet
    preconditions for the mentioned action; If the plan is
    executable but not goal reaching then include (2)
    "unmet_goals": A list of unmet goal conditions in the JSON.
    Finally include (3) "valid": a binary value that tells if the
    plan is valid or not i.e., the plan when executed satisfies the
    goal conditions. Include only one of the keys (1) or (2) based
    on the type of plan invalidity.
Let's think step by step
[THOUGHTS]
```

Prompt To Elicit Verification (Swapping Answer and Reason Order)

```
I am playing with a set of blocks where I need to arrange the
    blocks into stacks. Here are the actions I can do

Pick up a block
Unstack a block from on top of another block
Put down a block
Stack a block on top of another block

I have the following restrictions on my actions:
I can only pick up or unstack one block at a time.
I can only pick up or unstack a block if my hand is empty.
I can only pick up a block if the block is on the table and the
    block is clear. A block is clear if the block has no other
    blocks on top of it and if the block is not picked up.
I can only unstack a block from on top of another block if the
    block I am unstacking was really on top of the other block.
I can only unstack a block from on top of another block if the
    block I am unstacking is clear.
Once I pick up or unstack a block, I am holding the block.
I can only put down a block that I am holding.
I can only stack a block on top of another block if I am holding
    the block being stacked.
I can only stack a block on top of another block if the block onto
    which I am stacking the block is clear.
Once I put down or stack a block, my hand becomes empty.
Once you stack a block on top of a second block, the second block
    is no longer clear.

[STATEMENT]
As initial conditions I have that, the red block is clear, the
    yellow block is clear, the hand is empty, the red block is on
    top of the blue block, the yellow block is on top of the orange
    block, the blue block is on the table and the orange block is
    on the table.
```

```
My goal is to have that the orange block is on top of the red
    block.
My plan is as follows:

[PLAN]
unstack the yellow block from on top of the orange block
put down the yellow block
pick up the orange block
stack the orange block on top of the red block
[PLAN END]

Verify whether the above plan is valid. Provide a JSON between
    tags [JSON] and [JSON_END] for the verification information.
    The JSON should contain three main keys: If the plan is invalid
    and inexecutable then include (1) "unmet_preconditions": This
    contains two more keys; (1.1) "action": This is the name of the
    first action that renders the plan inexecutable (1.2)
    "preconditions": A list of unmet preconditions for the
    mentioned action; If the plan is executable but not goal
    reaching then include (2) "unmet_goals": A list of unmet goal
    conditions in the JSON. Finally include (3) "valid": a binary
    value that tells if the plan is valid or not i.e., the plan
    when executed satisfies the goal conditions. Include only one
    of the keys (1) or (2) based on the type of plan invalidity.
```

### A.5.3  BACKPROMPTS - BLOCKSWORLD

**Backprompt Generated From Self-Critique**

```
Plan is invalid.
The plan fails because after you stack the blue block on top of
    the red block, you cannot unstack the yellow block from on top
    of the orange block because the yellow block is not clear. It
    has the blue block on top of it.
```

**Backprompt Generated From Sound Verifier (First)**

```
The above plan is invalid.
The following action at step 7 has an unmet precondition:
pick up the red block
The unmet precondition is:
the red block is clear
```

**Backprompt Generated From Sound Verifier (Full)**

```
The above plan is invalid.
There are unsatisfied preconditions.
The following action at step 7 has an unmet precondition:
pick up the red block
The unmet precondition is:
the red block is clear
The following action at step 8 has an unmet precondition:
stack the red block on top of the orange block
The unmet precondition is:
the orange block is clear
The following action at step 11 has an unmet precondition:
pick up the yellow block
The unmet precondition is:
the yellow block is on the table
```

### A.5.4 COMPLETE EXAMPLE OF ITERATIVE PROMPT SEQUENCE - BLOCKSWORLD

---

**Full Sequence of (LLM-Generated) Backprompts**

```
I am playing with a set of blocks where I need to arrange the
    blocks into stacks. Here are the actions I can do

Pick up a block
Unstack a block from on top of another block
Put down a block
Stack a block on top of another block

I have the following restrictions on my actions:
I can only pick up or unstack one block at a time.
I can only pick up or unstack a block if my hand is empty.
I can only pick up a block if the block is on the table and the
    block is clear. A block is clear if the block has no other
    blocks on top of it and if the block is not picked up.
I can only unstack a block from on top of another block if the
    block I am unstacking was really on top of the other block.
I can only unstack a block from on top of another block if the
    block I am unstacking is clear.
Once I pick up or unstack a block, I am holding the block.
I can only put down a block that I am holding.
I can only stack a block on top of another block if I am holding
    the block being stacked.
I can only stack a block on top of another block if the block onto
    which I am stacking the block is clear.
Once I put down or stack a block, my hand becomes empty.
Once you stack a block on top of a second block, the second block
    is no longer clear.

[STATEMENT]
As initial conditions I have that, the red block is clear, the
    blue block is clear, the yellow block is clear, the hand is
    empty, the blue block is on top of the orange block, the red
    block is on the table, the orange block is on the table and the
    yellow block is on the table.
My goal is to have that the red block is on top of the blue block,
    the blue block is on top of the yellow block and the yellow
    block is on top of the orange block.

My plan is as follows:

[PLAN]
unstack the blue block from on top of the orange block
stack the blue block on top of the red block
pick up the yellow block
stack the yellow block on top of the orange block
unstack the blue block from on top of the red block
stack the blue block on top of the yellow block
pick up the red block
stack the red block on top of the blue block
[PLAN END]

[STATEMENT]
As initial conditions I have that, the blue block is clear, the
    orange block is clear, the hand is empty, the orange block is
    on top of the yellow block, the yellow block is on top of the
    red block, the red block is on the table and the blue block is
    on the table.
```

```
My goal is to have that the red block is on top of the orange
    block, the blue block is on top of the red block and the yellow
    block is on top of the blue block.

My plan is as follows:

[PLAN]
-----------GPT4-as-Planner Response-----------
unstack the orange block from on top of the yellow block
put down the orange block
unstack the yellow block from on top of the red block
stack the yellow block on top of the orange block
pick up the blue block
stack the blue block on top of the red block
unstack the yellow block from on top of the orange block
stack the yellow block on top of the blue block
pick up the red block
unstack the blue block from on top of the red block
stack the blue block on top of the yellow block
stack the red block on top of the orange block
[PLAN END]
-----------GPT4-as-Verifier Response-----------
Plan is invalid. The plan fails because after you stack the blue
    block on top of the red block, you cannot unstack the yellow
    block from on top of the orange block because the yellow block
    is not clear. It has the blue block on top of it.
-----------GPT4-as-Planner Response-----------
I apologize for the oversight. Here is the corrected plan:

[PLAN]
unstack the orange block from on top of the yellow block
put down the orange block
unstack the yellow block from on top of the red block
put down the yellow block
pick up the blue block
stack the blue block on top of the red block
pick up the orange block
stack the orange block on top of the red block
pick up the yellow block
stack the yellow block on top of the blue block
[PLAN END]
-----------GPT4-as-Verifier Response-----------
Plan is invalid. The plan fails because after stacking the blue
    block on top of the red block, you are trying to stack the
    orange block on top of the red block. However, the red block is
    not clear because the blue block is on top of it.
-----------GPT4-as-Planner Response-----------
I apologize for the mistake. Here is the corrected plan:

[PLAN]
unstack the orange block from on top of the yellow block
put down the orange block
unstack the yellow block from on top of the red block
put down the yellow block
pick up the blue block
stack the blue block on top of the red block
pick up the orange block
stack the orange block on top of the blue block
pick up the yellow block
stack the yellow block on top of the orange block
[PLAN END]
-----------GPT4-as-Verifier Response-----------
Plan is valid.
```

### A.5.5 LLM AS VERIFIER - BLOCKSWORLD

---

**Examples of LLM Output on the Verification Task**

```
[STATEMENT]
As initial conditions I have that, the blue block is clear, the
    orange block is clear, the hand is empty, the orange block is
    on top of the yellow block, the yellow block is on top of the
    red block, the red block is on the table and the blue block is
    on the table.
My goal is to have that the red block is on top of the orange
    block, the blue block is on top of the red block and the yellow
    block is on top of the blue block.

My plan is as follows:

[PLAN]
------------GPT-4 Response-------------
unstack the orange block from on top of the yellow block
put down the orange block
unstack the yellow block from on top of the red block
stack the yellow block on top of the orange block
pick up the blue block
stack the blue block on top of the red block
unstack the yellow block from on top of the orange block
stack the yellow block on top of the blue block
pick up the red block
unstack the blue block from on top of the red block
stack the blue block on top of the yellow block
stack the red block on top of the orange block
[PLAN END]
---------GPT-4 Verifier Incorrect Response---------
Plan is invalid. The plan fails because after you stack the blue
    block on top of the red block, you cannot unstack the yellow
    block from on top of the orange block because the yellow block
    is not clear. It has the blue block on top of it.
```

---

### A.5.6 PROMPTS - MYSTERY BLOCKSWORLD

All of following examples are built on the same graph instance.

---

**Baseline, Direct Prompt**

```
I am playing with a set of objects. Here are the actions I can do

   Attack object
   Feast object from another object
   Succumb object
   Overcome object from another object

I have the following restrictions on my actions:
    To perform Attack action, the following facts need to be true:
   Province object, Planet object, Harmony.
    Once Attack action is performed the following facts will be
   true: Pain object.
    Once Attack action is performed the following facts will be
   false: Province object, Planet object, Harmony.
    To perform Succumb action, the following facts need to be
   true: Pain object.
    Once Succumb action is performed the following facts will be
   true: Province object, Planet object, Harmony.
```

---

```
    Once Succumb action is performed the following facts will be
    false: Pain object.
    To perform Overcome action, the following needs to be true:
    Province other object, Pain object.
    Once Overcome action is performed the following will be true:
    Harmony, Province object, Object Craves other object.
    Once Overcome action is performed the following will be false:
    Province other object, Pain object.
    To perform Feast action, the following needs to be true:
    Object Craves other object, Province object, Harmony.
    Once Feast action is performed the following will be true:
    Pain object, Province other object.
    Once Feast action is performed the following will be false:,
    Object Craves other object, Province object, Harmony.

[STATEMENT]
As initial conditions I have that, object a craves object b,
    object d craves object c, harmony, planet object b, planet
    object c, province object a and province object d.
My goal is to have that object c craves object a.

My plan is as follows:

[PLAN]
feast object d from object c
succumb object d
attack object c
overcome object c from object a
[PLAN END]

[STATEMENT]
As initial conditions I have that, object b craves object c,
    object c craves object d, object d craves object a, harmony,
    planet object a and province object b.
My goal is to have that object a craves object c and object d
    craves object a.

My plan is as follows:

[PLAN]
```

### Example LLM Response

```
feast object b from object c
succumb object b
attack object a
overcome object a from object c
feast object d from object a
succumb object d
attack object c
overcome object c from object a
[PLAN END]
```

### Prompt To Elicit Verification (Open Ended)

```
I am playing with a set of objects. Here are the actions I can do

    Attack object
    Feast object from another object
    Succumb object
```

```
   Overcome object from another object

I have the following restrictions on my actions:
    To perform Attack action, the following facts need to be true:
    Province object, Planet object, Harmony.
    Once Attack action is performed the following facts will be
    true: Pain object.
    Once Attack action is performed the following facts will be
    false: Province object, Planet object, Harmony.
    To perform Succumb action, the following facts need to be
    true: Pain object.
    Once Succumb action is performed the following facts will be
    true: Province object, Planet object, Harmony.
    Once Succumb action is performed the following facts will be
    false: Pain object.
    To perform Overcome action, the following needs to be true:
    Province other object, Pain object.
    Once Overcome action is performed the following will be true:
    Harmony, Province object, Object Craves other object.
    Once Overcome action is performed the following will be false:
    Province other object, Pain object.
    To perform Feast action, the following needs to be true:
    Object Craves other object, Province object, Harmony.
    Once Feast action is performed the following will be true:
    Pain object, Province other object.
    Once Feast action is performed the following will be false:,
    Object Craves other object, Province object, Harmony.

[STATEMENT]
As initial conditions I have that, object b craves object c,
    object c craves object d, object d craves object a, harmony,
    planet object a and province object b.
My goal is to have that object a craves object c and object d
    craves object a.
My plan is as follows:

[PLAN]
feast object a from object d
succumb object a
attack object a
overcome object a from object c
feast object c from object d
succumb object c
attack object c
overcome object c from object a
[PLAN END]

Verify whether the above plan is valid. If it is valid, please say
    "Plan is valid." and nothing else. If it is invalid, please say
    "Plan is invalid." and then provide feedback on why the plan
    fails.
```

---

**Prompt To Elicit Verification (Format based)**

```
I am playing with a set of objects. Here are the actions I can do

   Attack object
   Feast object from another object
   Succumb object
   Overcome object from another object
```

```
I have the following restrictions on my actions:
    To perform Attack action, the following facts need to be true:
    Province object, Planet object, Harmony.
    Once Attack action is performed the following facts will be
    true: Pain object.
    Once Attack action is performed the following facts will be
    false: Province object, Planet object, Harmony.
    To perform Succumb action, the following facts need to be
    true: Pain object.
    Once Succumb action is performed the following facts will be
    true: Province object, Planet object, Harmony.
    Once Succumb action is performed the following facts will be
    false: Pain object.
    To perform Overcome action, the following needs to be true:
    Province other object, Pain object.
    Once Overcome action is performed the following will be true:
    Harmony, Province object, Object Craves other object.
    Once Overcome action is performed the following will be false:
    Province other object, Pain object.
    To perform Feast action, the following needs to be true:
    Object Craves other object, Province object, Harmony.
    Once Feast action is performed the following will be true:
    Pain object, Province other object.
    Once Feast action is performed the following will be false:,
    Object Craves other object, Province object, Harmony.

[STATEMENT]
As initial conditions I have that, object a craves object b,
    object d craves object c, harmony, planet object b, planet
    object c, province object a and province object d.
My goal is to have that object c craves object a.
My plan is as follows:

[PLAN]
feast object d from object c
succumb object d
attack object c
overcome object c from object a
[PLAN END]

Verify whether the above plan is valid. Provide a JSON between
    tags [JSON] and [JSON\_END] for the verification information.
    The JSON should contain three main keys: (1) "valid": a binary
    value that tells if the plan is valid or not i.e., the plan
    when executed satisfies the goal conditions. If the plan is
    invalid and inexecutable then include (2)
    "unmet\_preconditions": This contains two more keys; (2.1)
    "action": This is the name of the first action that renders the
    plan inexecutable (2.2) "preconditions": A list of unmet
    preconditions for the mentioned action; If the plan is
    executable but not goal reaching then include (3)
    "unmet\_goals": A list of unmet goal conditions in the JSON.
    Include only one of the keys (2) or (3) based on the plan
    invalidity.
```

## Prompt To Elicit Verification (Chain of thought based)

```
I am playing with a set of objects. Here are the actions I can do

    Attack object
    Feast object from another object
```

```
   Succumb object
   Overcome object from another object

I have the following restrictions on my actions:
    To perform Attack action, the following facts need to be true:
    Province object, Planet object, Harmony.
    Once Attack action is performed the following facts will be
    true: Pain object.
    Once Attack action is performed the following facts will be
    false: Province object, Planet object, Harmony.
    To perform Succumb action, the following facts need to be
    true: Pain object.
    Once Succumb action is performed the following facts will be
    true: Province object, Planet object, Harmony.
    Once Succumb action is performed the following facts will be
    false: Pain object.
    To perform Overcome action, the following needs to be true:
    Province other object, Pain object.
    Once Overcome action is performed the following will be true:
    Harmony, Province object, Object Craves other object.
    Once Overcome action is performed the following will be false:
    Province other object, Pain object.
    To perform Feast action, the following needs to be true:
    Object Craves other object, Province object, Harmony.
    Once Feast action is performed the following will be true:
    Pain object, Province other object.
    Once Feast action is performed the following will be false:,
    Object Craves other object, Province object, Harmony.

[STATEMENT]
As initial conditions I have that, object a craves object b,
    object d craves object c, harmony, planet object b, planet
    object c, province object a and province object d.
My goal is to have that object c craves object a.
My plan is as follows:

[PLAN]
feast object d from object c
succumb object d
attack object c
overcome object c from object a
[PLAN END]

Verify whether the above plan is valid. You will think step by
    step and output intermediate reasoning steps and thoughts for
    the verification after the [THOUGHTS] tag. Then, provide a JSON
    between the tags [JSON] and [JSON_END] for the verification
    information. The JSON should contain three main keys: If the
    plan is invalid and inexecutable then include (1)
    "unmet_preconditions": This contains two more keys; (1.1)
    "action": This is the name of the first action that renders the
    plan inexecutable (1.2) "preconditions": A list of unmet
    preconditions for the mentioned action; If the plan is
    executable but not goal reaching then include (2)
    "unmet_goals": A list of unmet goal conditions in the JSON.
    Finally include (3) "valid": a binary value that tells if the
    plan is valid or not i.e., the plan when executed satisfies the
    goal conditions. Include only one of the keys (1) or (2) based
    on the type of plan invalidity.
Let's think step by step
[THOUGHTS]
```

---

**Prompt To Elicit Verification (Swapping Answer and Reason Order)**

```
I am playing with a set of objects. Here are the actions I can do

   Attack object
   Feast object from another object
   Succumb object
   Overcome object from another object

I have the following restrictions on my actions:
    To perform Attack action, the following facts need to be true:
    Province object, Planet object, Harmony.
    Once Attack action is performed the following facts will be
    true: Pain object.
    Once Attack action is performed the following facts will be
    false: Province object, Planet object, Harmony.
    To perform Succumb action, the following facts need to be
    true: Pain object.
    Once Succumb action is performed the following facts will be
    true: Province object, Planet object, Harmony.
    Once Succumb action is performed the following facts will be
    false: Pain object.
    To perform Overcome action, the following needs to be true:
    Province other object, Pain object.
    Once Overcome action is performed the following will be true:
    Harmony, Province object, Object Craves other object.
    Once Overcome action is performed the following will be false:
    Province other object, Pain object.
    To perform Feast action, the following needs to be true:
    Object Craves other object, Province object, Harmony.
    Once Feast action is performed the following will be true:
    Pain object, Province other object.
    Once Feast action is performed the following will be false:,
    Object Craves other object, Province object, Harmony.

[STATEMENT]
As initial conditions I have that, object a craves object b,
    object d craves object c, harmony, planet object b, planet
    object c, province object a and province object d.
My goal is to have that object c craves object a.
My plan is as follows:

[PLAN]
feast object d from object c
succumb object d
attack object c
overcome object c from object a
[PLAN END]

Verify whether the above plan is valid. Provide a JSON between
    tags [JSON] and [JSON_END] for the verification information.
    The JSON should contain three main keys: If the plan is invalid
    and inexecutable then include (1) "unmet_preconditions": This
    contains two more keys; (1.1) "action": This is the name of the
    first action that renders the plan inexecutable (1.2)
    "preconditions": A list of unmet preconditions for the
    mentioned action; If the plan is executable but not goal
    reaching then include (2) "unmet_goals": A list of unmet goal
    conditions in the JSON. Finally include (3) "valid": a binary
    value that tells if the plan is valid or not i.e., the plan
    when executed satisfies the goal conditions. Include only one
    of the keys (1) or (2) based on the type of plan invalidity.
```

### A.5.7 BACKPROMPTS - MYSTERY BLOCKSWORLD

---

**Backprompt Generated From Self-Critique**

```
Plan is invalid.

The first action "feast object b from object c" is not possible
    because the initial conditions do not include "Province object
    b" which is required for the Feast action.
```

---

**Backprompt Generated From Sound Verifier (First)**

```
The above plan is invalid.
The following action at step 3 has an unmet precondition:
attack object a
The unmet precondition is:
province object a
```

---

**Backprompt Generated From Sound Verifier (Full)**

```
The above plan is invalid.
There is an unmet goal condition. This is:
object d craves object a
There are unsatisfied preconditions.
The following action at step 3 has an unmet precondition:
attack object a
The unmet precondition is:
province object a
The following action at step 5 has an unmet precondition:
feast object d from object a
The unmet precondition is:
province object d
```

---

### A.5.8 COMPLETE EXAMPLE OF ITERATIVE PROMPT SEQUENCE - MYSTERY BLOCKSWORLD

---

**Full Sequence of 15 (LLM-Generated) Backprompts**

```
I am playing with a set of objects. Here are the actions I can do

    Attack object
    Feast object from another object
    Succumb object
    Overcome object from another object

I have the following restrictions on my actions:
     To perform Attack action, the following facts need to be true:
    Province object, Planet object, Harmony.
     Once Attack action is performed the following facts will be
    true: Pain object.
     Once Attack action is performed the following facts will be
    false: Province object, Planet object, Harmony.
     To perform Succumb action, the following facts need to be
    true: Pain object.
     Once Succumb action is performed the following facts will be
    true: Province object, Planet object, Harmony.
     Once Succumb action is performed the following facts will be
    false: Pain object.
     To perform Overcome action, the following needs to be true:
    Province other object, Pain object.
```

```
     Once Overcome action is performed the following will be true:
     Harmony, Province object, Object Craves other object.
     Once Overcome action is performed the following will be false:
     Province other object, Pain object.
     To perform Feast action, the following needs to be true:
     Object Craves other object, Province object, Harmony.
     Once Feast action is performed the following will be true:
     Pain object, Province other object.
     Once Feast action is performed the following will be false:,
     Object Craves other object, Province object, Harmony.

[STATEMENT]
As initial conditions I have that, object a craves object b,
     object d craves object c, harmony, planet object b, planet
     object c, province object a and province object d.
My goal is to have that object c craves object a.

My plan is as follows:

[PLAN]
feast object d from object c
succumb object d
attack object c
overcome object c from object a
[PLAN END]

[STATEMENT]
As initial conditions I have that, object b craves object c,
     object c craves object d, object d craves object a, harmony,
     planet object a and province object b.
My goal is to have that object a craves object c and object d
     craves object a.

My plan is as follows:

[PLAN]
-----------GPT4-as-Planner Response-----------
feast object b from object c
succumb object b
attack object a
overcome object a from object c
feast object d from object a
succumb object d
attack object c
overcome object c from object a
[PLAN END]
-----------GPT4-as-Verifier Response-----------
Plan is invalid.

The first action "feast object b from object c" is not possible
     because the initial conditions do not include "Province object
     b" which is required for the Feast action.
-----------GPT4-as-Planner Response-----------
I apologize for the oversight. Let's correct the plan.

[PLAN]
attack object a
succumb object a
feast object b from object c
overcome object c from object a
attack object d
succumb object d
feast object c from object a
```

```
overcome object a from object c
[PLAN END]
----------GPT4-as-Verifier Response----------
Plan is invalid.

The first action "attack object a" cannot be performed because the
    conditions for the attack action are not met. According to the
    restrictions, to perform an attack action, the following facts
    need to be true: Province object, Planet object, Harmony. In
    the initial conditions, you have Province object b, not
    Province object a. Therefore, the attack action cannot be
    performed on object a.
----------GPT4-as-Planner Response----------
I apologize for the confusion. Let's revise the plan.

[PLAN]
attack object b
succumb object b
feast object c from object d
overcome object d from object a
attack object a
succumb object a
feast object d from object a
overcome object a from object c
[PLAN END]
----------GPT4-as-Verifier Response----------
Plan is invalid.

The first action "attack object b" cannot be performed because the
    conditions for the attack action are not met. The attack action
    requires the facts: Province object, Planet object, Harmony to
    be true. In the initial conditions, you have Province object b
    and Harmony, but you do not have Planet object. Therefore, the
    attack action cannot be performed.
----------GPT4-as-Planner Response----------
I apologize for the confusion. Let's revise the plan.

[PLAN]
feast object b from object c
succumb object b
attack object d
overcome object d from object a
feast object a from object d
succumb object a
attack object c
overcome object c from object a
[PLAN END]
----------GPT4-as-Verifier Response----------
Plan is invalid.

The first action "feast object b from object c" cannot be
    performed because the initial conditions do not satisfy the
    requirements for the Feast action. According to the rules, to
    perform Feast action, the following needs to be true: Object
    Craves other object, Province object, Harmony. In the initial
    conditions, object b craves object c, harmony, and province
    object b are true. However, the Feast action requires that the
    object performing the action (object b in this case) is the
    Province object, which is not the case in the initial
    conditions. Therefore, the first action cannot be performed,
    making the plan invalid.
----------GPT4-as-Planner Response----------
I apologize for the confusion. Let's revise the plan.
```

```
[PLAN]
attack object b
succumb object b
feast object c from object d
overcome object d from object a
feast object a from object d
succumb object a
attack object c
overcome object c from object a
[PLAN END]
-----------GPT4-as-Verifier Response-----------
Plan is invalid.

The first action "attack object b" is valid because the initial
    conditions meet the requirements for the attack action.
    However, the second action "succumb object b" is invalid
    because the "Pain object" fact is not true after the first
    action. The "Attack object b" action makes the "Pain object"
    fact true, but it also makes the "Province object" fact false,
    which is a requirement for the "Succumb" action. Therefore, the
    plan fails at the second action.
-----------GPT4-as-Planner Response-----------
I apologize for the confusion. Let's revise the plan.

[PLAN]
attack object b
succumb object b
feast object c from object d
overcome object d from object a
feast object a from object d
succumb object a
attack object c
overcome object c from object a
[PLAN END]
-----------GPT4-as-Verifier Response-----------
Plan is invalid.

The first action "attack object b" is valid because the initial
    conditions meet the requirements for the attack action:
    Province object, Planet object, Harmony.

However, the second action "succumb object b" is invalid. The
    succumb action requires the fact "Pain object" to be true,
    which is achieved after the attack action. But the object for
    the succumb action should be the same as the object that was
    attacked, which is not the case in your plan.

The rest of the plan cannot be validated because it depends on the
    outcome of the invalid succumb action.
-----------GPT4-as-Planner Response-----------
I apologize for the confusion. Let's revise the plan.

[PLAN]
attack object b
succumb object b
feast object b from object c
overcome object c from object a
attack object a
succumb object a
feast object a from object d
overcome object d from object a
[PLAN END]
```

```
-----------GPT4-as-Verifier Response-----------
Plan is invalid.

The plan fails at the first step "attack object b". According to
    the rules, to perform the Attack action, the following facts
    need to be true: Province object, Planet object, Harmony. In
    the initial conditions, you have Province object b, Planet
    object a, and Harmony. However, for the Attack action, you need
    the same object for Province and Planet, which is not the case
    here. Therefore, you cannot perform the Attack action on object
    b.
-----------GPT4-as-Planner Response-----------
I apologize for the confusion. Let's revise the plan.

[PLAN]
attack object a
succumb object a
feast object b from object c
overcome object c from object a
attack object a
succumb object a
feast object a from object d
overcome object d from object a
[PLAN END]
-----------GPT4-as-Verifier Response-----------
Plan is invalid.

The first action "attack object a" cannot be performed because the
    initial conditions do not meet the requirements for the attack
    action. The attack action requires the facts: Province object,
    Planet object, Harmony to be true. In the initial conditions,
    the Province object is object b, not object a. Therefore, the
    attack action cannot be performed on object a.
-----------GPT4-as-Planner Response-----------
I apologize for the confusion. Let's revise the plan.

[PLAN]
attack object d
succumb object d
feast object a from object b
overcome object b from object c
attack object c
succumb object c
feast object c from object d
overcome object d from object a
[PLAN END]
-----------GPT4-as-Verifier Response-----------
Plan is invalid.

The first action "attack object d" cannot be performed because the
    conditions for the Attack action are not met. According to the
    rules, to perform the Attack action, the following facts need
    to be true: Province object, Planet object, Harmony. In the
    initial conditions, you have Province object b, Planet object
    a, and Harmony. However, you are trying to attack object d, not
    object a or b. Therefore, the first action in your plan is not
    valid.
-----------GPT4-as-Planner Response-----------
I apologize for the confusion. Let's revise the plan.

[PLAN]
attack object b
succumb object b
```

```
feast object b from object c
overcome object c from object a
attack object a
succumb object a
feast object a from object d
overcome object d from object a
[PLAN END]
-----------GPT4-as-Verifier Response-----------
Plan is invalid.

The first action "attack object b" is invalid because the
    conditions for the attack action are not met. The conditions
    for the attack action are: Province object, Planet object,
    Harmony. In the initial conditions, you have Province object b,
    Planet object a, and Harmony. The Province object and Planet
    object must be the same for the attack action to be valid.
    Therefore, you cannot attack object b with the given initial
    conditions.
-----------GPT4-as-Planner Response-----------
I apologize for the confusion. Let's revise the plan.

[PLAN]
feast object a from object b
succumb object a
attack object a
overcome object a from object c
feast object c from object d
succumb object c
attack object c
overcome object c from object a
[PLAN END]
-----------GPT4-as-Verifier Response-----------
Plan is invalid.

The first action "feast object a from object b" is not possible
    because the initial conditions state that "object b craves
    object c", not "object b craves object a". Therefore, the Feast
    action cannot be performed as the condition "Object Craves
    other object" is not met.

The plan fails at the first step.
-----------GPT4-as-Planner Response-----------
I apologize for the confusion. Let's revise the plan.

[PLAN]
feast object d from object a
succumb object d
attack object d
overcome object d from object c
feast object c from object d
succumb object c
attack object c
overcome object c from object a
[PLAN END]
-----------GPT4-as-Verifier Response-----------
Plan is invalid.

The first action "feast object d from object a" cannot be
    performed because the initial conditions do not satisfy the
    requirements for the Feast action. The Feast action requires
    that the object performing the action (in this case, object d)
    craves the other object (in this case, object a). However, the
    initial conditions state that object d craves object a, not the
```

```
    other way around. Therefore, the Feast action cannot be
    performed, and the plan is invalid.
-----------GPT4-as-Planner Response-----------
I apologize for the confusion. Let's revise the plan.

[PLAN]
feast object a from object b
succumb object a
attack object a
overcome object a from object c
feast object c from object d
succumb object c
attack object c
overcome object c from object a
[PLAN END]
-----------GPT4-as-Verifier Response-----------
Plan is invalid.

The first action "feast object a from object b" cannot be
    performed because the initial conditions state that "object b
    craves object c", not "object a". Therefore, the Feast action
    cannot be performed as it requires the fact "Object Craves
    other object" to be true, which in this case would be "object b
    craves object a".
-----------GPT4-as-Planner Response-----------
I apologize for the confusion. Let's revise the plan.

[PLAN]
feast object d from object c
succumb object d
attack object d
overcome object d from object a
feast object a from object d
succumb object a
attack object a
overcome object a from object c
[PLAN END]
-----------GPT4-as-Verifier Response-----------
Plan is invalid.

The first action in your plan, "feast object d from object c",
    cannot be performed because the initial conditions do not
    include "Object d craves object c". The initial conditions
    state that "object b craves object c, object c craves object d,
    object d craves object a". Therefore, the first action in your
    plan is not valid according to the restrictions on the Feast
    action.
-----------GPT4-as-Planner Response-----------
I apologize for the confusion. Let's revise the plan.

[PLAN]
feast object a from object d
succumb object a
attack object a
overcome object a from object c
feast object c from object d
succumb object c
attack object c
overcome object c from object a
[PLAN END]
-----------GPT4-as-Verifier Response-----------
Plan is invalid.
```

```
The first action in your plan, "feast object a from object d",
    cannot be performed because the initial conditions do not
    include "Object a craves object d". According to the rules, to
    perform the Feast action, the following needs to be true:
    "Object Craves other object, Province object, Harmony". In your
    initial conditions, you have "Object b craves object c, Object
    c craves object d, Object d craves object a, Harmony, Planet
    object a and Province object b". Therefore, you cannot perform
    the Feast action on object a from object d as your first action.
```

### A.5.9   LLM AS VERIFIER - MYSTERY BLOCKSWORLD

**Examples of LLM Output on the Verification Task**

```
=========Current State==========
object b craves object c, object c craves object d, object d
    craves object a, harmony, planet object a and province object b.

==========GPT-4-as-Verifier============
Plan is invalid.

The first action "feast object b from object c" is not possible
    because the initial conditions do not include "Province object
    b" which is required for the Feast action.
==========GPT-4-as-Verifier============
Plan is invalid.

The first action "feast object b from object c" cannot be
    performed because the initial conditions do not satisfy the
    requirements for the Feast action. According to the rules, to
    perform Feast action, the following needs to be true: Object
    Craves other object, Province object, Harmony. In the initial
    conditions, object b craves object c, harmony, and province
    object b are true. However, the Feast action requires that the
    object performing the action (object b in this case) is the
    Province object, which is not the case in the initial
    conditions. Therefore, the first action cannot be performed,
    making the plan invalid.

==========GPT-4-as-Verifier============
Plan is invalid.

The first action "attack object b" is valid because the initial
    conditions meet the requirements for the attack action:
    Province object, Planet object, Harmony.

However, the second action "succumb object b" is invalid. The
    succumb action requires the fact "Pain object" to be true,
    which is achieved after the attack action. But the object for
    the succumb action should be the same as the object that was
    attacked, which is not the case in your plan.

The rest of the plan cannot be validated because it depends on the
    outcome of the invalid succumb action.
```

