# OpenReview forum: "On the self-verification limitations of large language models on reasoning and planning tasks"
_ICLR.cc/2025/Conference — ICLR 2025 Poster_

### Official Review · Reviewer_jfRh · 2024-10-17

**Soundness:** 3
**Presentation:** 2
**Contribution:** 3
**Rating:** 6
**Confidence:** 4

**Summary:**

This paper mainly evaluates the self-critique and verification capabilities of LLMs in comparison to sound verifiers. It concludes that the self-verification loop conducted by LLMs is not as helpful as previous works claimed. On the contrary, it might hurt LLM performance in some planning domains.

**Strengths:**

This paper has the following strengths.

Originality:
This paper systematically evaluates LLMs' self-verification capability by careful ablation and comparison to symbolic processors. It finds that LLM verification is not very helpful in some planning cases. This is novel and interesting.

Quality:
This paper considers problems of memorization and lack of ground truth during evaluation, which are important concerns.

Clarity:
The experiment setup is quite clear.

Significance:
Knowing what LLMs are actually doing when people claim they can do a lot of things is very important.

**Weaknesses:**

1. Is it fair to compare LLMs with oracle verifiers? The finding that LLM self-critique can sometimes downgrade their generation performance is interesting, but oracle processors are not always accessible in all tasks (as the authors mentioned in their paper, some tasks such as creative writing do not have a ground truth answer). I'm not surprised that oracle verifier is improving LLM performance, but I wonder if it is possible that LLM can serve as a decent alternative when sound verifier is absent in general areas?

2. The task domains are constrained while the general conclusion is very strong. The authors evaluated three main planning tasks and one LLM (GPT-4) with four datasets, with no clear objective as to why the conclusion drawn from these three specific tasks can be generalized. It is unclear why these tasks and datasets can represent general LLMs' lack of self-verification capability in a wide ranges of tasks.

3. Some other details in this paper are also overclaimed, e.g., on page 2, the authors claim "...the state-of-the-art GPT-4" while GPT-4 is not SotA in many benchmarks anymore (in [1], inter alia).

4. There are quite some typos. Writing needs to be done with more care.

[1] Dubey, A., Jauhri, A., Pandey, A., Kadian, A., Al-Dahle, A., Letman, A., ... & Ganapathy, R. (2024). The llama 3 herd of models. arXiv preprint arXiv:2407.21783.

**Questions:**

I don't have more questions but authors are encouraged to address my concerns in the weakness section.

---

> ### Author Response · Authors · 2024-11-22
>
> We sincerely appreciate reviewer jfRh for their feedback. Below, we provide clarifications in response to the major concerns raised in the review.
>
> **Choice of domains:**
>
> We would like to first point out that our work’s primary aim is to caution the community on using LLM self-verification methods for reasoning tasks. As mentioned in our related work section, we believe that reasoning is a fraught term. In previous work it has been unclear as to what definition of reasoning the works presuppose when making claims about LLM reasoning capabilities. Due to these shifting definitions and implicit assumptions, making concrete claims or pinning them down becomes a lot more difficult. This is why we restrict ourselves to focus on fully specified, verifiable problems that can be solved by deductive methods and have oracle verifiers. This allows us to check the quality of both binary verification and critique generated by the LLM. Furthermore, the algorithmic abilities these domains test are fundamental—any other reasoning task must include components that test these same capabilities, or else be only a retrieval task. We have made this explicit in the introduction section of the revised version of the paper.
>
> Our results show that using LLMs as verifiers and within a self-verification loop is harmful in general and much more care should be taken when deploying LLMs in such a loop, specifically in reasoning tasks. Per our conclusion, we expect our results to be applicable to many overall ambiguous and complex tasks, because partial verifiers can be constructed and used together. Most real-world tasks contain a mixture of formally verifiable and tacit aspects. For example, see the TravelPlanning benchmark which combines “hard” and “soft” critics of LLM-generated plans. Or consider linters, unit tests, etc in software development.
>
> While we reported all our experiments on GPT4, our partial empirical studies with other LLMs (as shown in the table below), including GPT-4o, have largely been in line with the results we reported on GPT4. Nevertheless, we are currently in the process of replicating the experiments on LLaMA, and hope to have the results incorporated before the end of the discussion period, but certainly for the camera-ready version.
>
> | Model       | Domain                | S.P | LLM+LLM |
> |-------------|-----------------------|-----|---------|
> | GPT-4o      | Graph coloring        | 9%  | 0%      |
> | GPT-4o-mini | Graph Coloring        | 0%  | 0%      |
> | GPT-4o-mini | Graph Coloring (Easy) | 30% | 0%      |
> | GPT-4o-mini | Blocksworld | 29% | 4%      |
> | GPT-4o-mini | Mystery Blocksworld | 1% | 0%      |
>
> **Oracle verifiers**
>
> The oracle case is useful because it allows us to carefully and systematically ablate the self-verification system. Note that there are two components to the oracle case: the sound verification which is an external system that ensures that any output is guaranteed correct, and the sound feedback which can be passed back to the LLM in part or in whole. In section 5.2, we use this distinction to test how much of the increased improvement is due to the LLM itself conditioning on both the claim that the answer is incorrect and the feedback passed back. What we find is that it doesn’t seem to matter much. When we remove the feedback entirely, we retain most of the performance improvement. This further shows that previous claims that LLMs improve because they effectively take in feedback are misleading.

---

> > ### Comment · Reviewer_jfRh · 2024-11-29
> >
> > Thanks the authors for the clarifications. I have raised my score.

---

### Official Review · Reviewer_NPSN · 2024-11-02

**Soundness:** 4
**Presentation:** 3
**Contribution:** 3
**Rating:** 6
**Confidence:** 4

**Summary:**

This paper discusses the effect of self-verification in LLM reasoning and planning, challenging the assumption that verifying correctness is easier than generating solutions. Through experiments in three domains (Game of 24, Graph Coloring, and STRIPS planning), it finds that self-verification does not improve accuracy and can even reduce performance. In contrast, a "sound external verifier" significantly enhances accuracy, with simpler re-prompting methods maintaining most of these benefits.

**Strengths:**

1. The presentation and writing are clear.
2. The authors made adjustments to the test datasets to ensure their validity. For example, they generated new instances to prevent test set memorization and modified the task solution format to better evaluate the self-verification pipeline.
3. The systematic analysis of self-verification provides insights into the cases where the verification module can help improve LLM performance.

**Weaknesses:**

1. The experiment design is not entirely persuasive.
The authors attempt to challenge the assumption that verifying correctness is easier than generating solutions, suggesting that self-verification does not improve performance. However, task difficulty should be considered. Apart from Blocksworld, the accuracy in other tasks is quite low. According to Table 2, "LLM Verification Results," verification performance varies across tasks, especially in terms of the False Negative Rate (FNR). In Blocksworld, which has the lowest FNR, self-verification actually improves task accuracy from 40% to 55%. This suggests there are cases where verification is easier than generation and where self-verification contributes positively to task accuracy. More tasks should be added for more comprehensive results.

2. The authors use a “sound” verifier as a comparison, but it’s unsurprising that a ground-truth verifier significantly improves performance. With ground-truth verification, the model can eliminate incorrect answers and focus on sampling correct ones, or at least provide ground-truth information. Taking the first point into account, a more nuanced conclusion could be that in tasks where verification is easier than generation, self-verification helps; otherwise, it has no benefit or even harms performance. The improvement limit for self-verification is thus bounded by the effectiveness of the “sound” verifier.

3. The exact GPT version should be specified, and more models, particularly advanced ones, should be tested for comprehensive results.

**Questions:**

Please discuss the points mentioned in the weakness part.

---

> ### Author Response · Authors · 2024-11-22
>
> Thank you for your thoughtful comments and especially the recognition that our experimental study sheds considerable light on if and when self-verification can be helpful.
>
> Unfortunately however, the rest of the review suffers from two serious misunderstandings of the paper–that we bring up and clarify below.
>
> In weakness 1 you say:
>
>
> > **The authors attempt to challenge the assumption that verifying correctness is easier than generating solutions, suggesting that self-verification does not improve performance**
>
>
> We are afraid that this interpretation is off the mark. Our paper is not concerned with the question of whether or not verification is easier than generation. We specifically choose tasks on which we know that verification is easier than generation. Formalizing this in terms of complexity theory, across all these tasks, generation is in a higher complexity class than verification. Proposed solutions can be verified in polynomial time, but generating these solutions is NP-complete (graph coloring)  or even PSPACE-complete (planning).
>
> What we challenge instead is the prevailing wisdom that LLMs are somehow sensitive to the lower computational complexity of verification. We argue that there is no reason to believe that LLMs will be better at verification than generation. Specifically,  we challenge the claim that in these scenarios LLMs can verify proposed solutions better than they can generate them, and thus robustly bootstrap their performance to higher levels.
>
> Our results indeed validate our belief/hypothesis, and show that—in fact—the gains seen in previous studies may be worse than illusory: in many cases, we see performance worsen with LLM self-critiques.
>
> Continuing on, the reviewer says
>
>
> >  **more nuanced conclusion could be that in tasks where verification is easier than generation, self-verification helps**
>
>
> As you can see from the clarification above, the paper in fact showcases the opposite of the reviewer’s proposed conclusion!
>
>
> About the concern that we are testing on benchmarks where the LLM accuracy is not very high to begin with
>
> > **However, task difficulty should be considered. Apart from Blocksworld, the accuracy in other tasks is quite low.**
>
> We would argue that  in practice, self-verification is most useful when base performance is low, which reinforces the validity of our testing on tasks that range from 4 to 40% standard prompting accuracy. Some of the previous studies have focused on tasks on which LLM performance is already very high, sometimes as high as 98%--which makes the whole evaluation of effectiveness of self-verification a moot point.
>
> We specifically chose a set of tasks where the initial accuracy of LLMs is low to begin with, in order to gain more significant resolution in our analysis of how much the self-verification scheme does or does not improve. Note that,
>
>
> > **With ground-truth verification, the model can eliminate incorrect answers and focus on sampling correct ones, or at least provide ground-truth information.**
>
> Our paper also addresses this point. As we discuss, ground-truth verification is providing two things: 1) an external system that decides whether an answer is correct and thus outputs it only if it is guaranteed and 2) some feedback to the LLM in the form of a backprompt. The claim that the model is itself eliminating incorrect answers is also tested—we compare ground truth verification with all, some, or even zero critique given in section 5.2. We find that, in fact, we can gain most of the performance improvement without providing anything about the previous wrong answers to the model! This indicates that the performance improvements seen are mainly from rejection sampling the model, rather than from any inherent LLM ability to “focus on sampling correct ones” when given feedback.

---

> ### Author Response · Authors · 2024-11-22
>
> > **The improvement limit for self-verification is thus bounded by the effectiveness of the “sound” verifier.**
>
> A sound verifier will, by definition, perfectly discriminate between correct and incorrect answers. The term “sound” here is borrowed from mathematical logic, where it essentially means that a system preserves truth. It is one of a desirable pair of properties for a system that together guarantee it will output all and only correct answers. The other property is completeness, which, in this case, would apply to the LLMs (are they capable of guessing all plausible solution candidates?). In general, such completeness is not guaranteed (as we discuss, approaches like ToT can be seen as externally inducing LLMs to generate a diverse set of candidate solutions through prompt diversification).
>
> Completeness would require that, for any problem of interest, the LLM would eventually output the correct answer (even if this answer were low probability and thus unlikely). However, even when we sample 150 answers per prompt in the case of Game of 24 (mentioned in A.2), the LLM only approaches 70% accuracy, and this accuracy asymptotes sharply. In short, we argue that it is not the effectiveness of the verifier that limits the improvement, but the incompleteness of the LLM.
>
> Finally, the issue of availability of sound verifiers is not as insurmountable as the reviewer seems to think it is. Most LLM verification systems are indeed tested on tasks where they already have external sound solvers/verifiers to provide synthetic data with correct verification labels.
>
> Furthermore, there are works in the literature (e.g. [1]) that show how verifiers can be teased out from the LLMs themselves with partial help of human or automated critics. Another idea is to learn verifiers from synthetic labeled data  and use them to verify solutions.
>
> The point of our work is to show that LLMs, out of the box, are no better at verification than generation.
>
> [1] Guan, Lin, Karthik Valmeekam, Sarath Sreedharan, and Subbarao Kambhampati. "Leveraging pre-trained large language models to construct and utilize world models for model-based task planning." Advances in Neural Information Processing Systems 36 (2023): 79081-79094.
>
> [2] Zhang, Lunjun, Arian Hosseini, Hritik Bansal, Mehran Kazemi, Aviral Kumar, and Rishabh Agarwal. "Generative verifiers: Reward modeling as next-token prediction." arXiv preprint arXiv:2408.15240 (2024).
>
> > **The exact GPT version should be specified and more models**
>
> Thank you for pointing this out. The exact model snapshot was GPT-4-0613. We have made this explicit in the revised version of the paper. While we reported all our experiments on GPT4, our partial empirical studies with other LLMs (as shown in the table below), including GPT-4o, have largely been in line with the results we reported on GPT4. Nevertheless, we are currently in the process of replicating the experiments on LLaMA, and hope to have the results incorporated before the end of the discussion period, but certainly for the camera-ready version.
>
> | Model       | Domain                | S.P | LLM+LLM |
> |-------------|-----------------------|-----|---------|
> | GPT-4o      | Graph coloring        | 9%  | 0%      |
> | GPT-4o-mini | Graph Coloring        | 0%  | 0%      |
> | GPT-4o-mini | Graph Coloring (Easy) | 30% | 0%      |
> | GPT-4o-mini | Blocksworld | 29% | 4%      |
> | GPT-4o-mini | Mystery Blocksworld | 1% | 0%      |

---

> > ### Comment · Reviewer_NPSN · 2024-11-23
> >
> > Thank you to the authors for their reply and extra experiments. Thank you for correcting my misunderstanding. Most of my concerns are resolved. I concur with the point that the paper is challenging and the conclusion that in many scenarios where verification is easier than generation, self-verification actually can't bootstrap the performance. I would like to offer a raise of the score. I still have some further questions to ask and discuss.
> >
> > 1. Task difficulty: I understand the authors selected tasks whose testing accuracy ranges from 4% to 40%, but according to the LLM+LLM results in Table 1, the only outlier is Blocksworld, whose LLM+LLM accuracy is larger than the S.P. accuracy. It also has the highest baseline accuracy (40%), while the others are significantly lower. The experiment would be more comprehensive if, in future studies not now, there are several tasks whose S.P. accuracy is around 50% and several around 80%.
> >
> > 2. Could the authors discuss more about why only Blocksworld is the outlier? Are there any reasons behind the numbers that make Blocksworld different from the other 3 tasks and lead to the difference in the performance of LLM+LLM? Insights into what kinds of tasks are feasible for self-critique would be valuable.
> >
> > 3. I concur that the selected tasks are easier to verify than to generate based on complexity theory. A minor question is, what kinds of tasks are harder to verify? Could the authors provide some examples?

---

> > > ### Author Response · Authors · 2024-11-27
> > >
> > > We thank the reviewer for their prompt response. Below we provide further clarifications to your questions.
> > >
> > > > **Could the authors discuss more about why only Blocksworld is the outlier? Are there any reasons behind the numbers that make Blocksworld different from the other 3 tasks and lead to the difference in the performance of LLM+LLM? Insights into what kinds of tasks are feasible for self-critique would be valuable.**
> > >
> > > There are a few things to note here:
> > >
> > > While Blocksworld is an outlier, this is likely due to it being a famous, well-represented domain in the pre-training data that has been studied since the 90s. However, this performance increase is much more brittle than it looks: Mystery Blocksworld is precisely a masked version of the exact same domain (we even use identical problems), and yet we don’t see the increase there. This bolsters the main point of our paper, because it shows that self-verification improvements can be illusory and not apply generally outside of test distributions. Furthermore, as these two domains are merely translations of each other, they retain the exact same properties from a complexity theoretic perspective, and so we see an even clearer example of how LLMs are not sensitive to this.
> > >
> > > This is part of the bigger message we are trying to get across: general verification is a form of reasoning. LLMs are not capable of reasoning--they can however do approximate retrieval based on their pretraining data. Thus they might look like they are reasoning on domains that are likely over-represented in their pretraining data but we see that on identical but slightly perturbed domains which are not well-represented, the performance collapses, and self-verification does not provide the kind of shortcuts previous work has hoped arises.
> > >
> > >
> > > > **I concur that the selected tasks are easier to verify than to generate based on complexity theory. A minor question is, what kinds of tasks are harder to verify? Could the authors provide some examples?**
> > >
> > > Obviously, computational complexity of verification cannot be higher than that of generation. So we interpret your question as "what are some cases where verification complexity is non-polynomial?" A canonical example is Satisfiability of Quantified Boolean Formulas (QBF). QBF can be seen as a generalization of Satisfiability--that underlies our Graph Coloring task. It can be used to express games such as GO and Reversi [1]. In the case of QBF, the satisfiability is P-Space complete, and verification of solution is NP-hard (in contrast to SAT and Graph Coloring, where the generationm is NP-Complete and verification is polynomial)
> > >
> > > In the case of planning too, while classical planning is P-Space complete for generation and polynomial for verification, more complex planning problems--such as conformant planning--where the agent doesn't have full observability--is known to be Exp-Space complete for generation and NP-Complete for verification [2]
> > >
> > > [1] Giunchiglia, Enrico, Paolo Marin, and Massimo Narizzano. "Reasoning with quantified boolean formulas." In Handbook of satisfiability, pp. 761-780. IOS Press, 2009.
> > >
> > > [2] Rintanen, Jussi. "Complexity of Planning with Partial Observability." In ICAPS, vol. 4, pp. 345-354. 2004.
> > >
> > > > **Task difficulty: I understand the authors selected tasks whose testing accuracy ranges from 4% to 40%, but according to the LLM+LLM results in Table 1, the only outlier is Blocksworld, whose LLM+LLM accuracy is larger than the S.P. accuracy. It also has the highest baseline accuracy (40%), while the others are significantly lower. The experiment would be more comprehensive if, in future studies not now, there are several tasks whose S.P. accuracy is around 50% and several around 80%.**
> > >
> > > We thank the reviewer for their suggestion. We believe that the major distinction in terms of accuracy would be how well-represented the reasoning domain could be in the pre-training data. We intend to investigate this in future work.

---

### Official Review · Reviewer_R3Bw · 2024-11-03

**Soundness:** 3
**Presentation:** 3
**Contribution:** 2
**Rating:** 8
**Confidence:** 4

**Summary:**

This is an experimental paper studying the the popular technique of self-verification of LLMs for enhancing apparent reasoning capabilities. The stringent experimental protocol shows that the self-verification does not really work and shows that other techniques like using formal (symbolic) verifiers are better suited to achieve automated reasoning with LLMs.

**Strengths:**

1) The paper is really well written and easy to to follow. The non-technical nature might have helped here.
2) I enjoyed reading the (extensive) related work section where problems in prior work are made explicit.
3) All claims made in the paper are substantiated by appropriate experiments.

**Weaknesses:**

1) For someone familiar with the field the findings might be a little bit obvious. However, given the amount of papers being published using self-verification, I nevertheless believe this to be an important study when it comes to combating the delusion of self-verification.

2) Given that this is an experimental study it might be good to also point out weaknesses in the experimental protocol. Specifically, making explicit all the assumptions that were made and what might change if they were not to hold. Concretely, what gives the authors the confidence that their findings have a high probability of standing the test of time.

**Questions:**

See point 2) in weaknesses and more specifically: are there any arguments that the findings do not only hold for the examined LLMs but in general for transformer-based auto-regressive models, or even other models like discrete diffusion models.


Minor comment:
The use of \citep and \citet is not correct. Also there are some small spelling/typesetting issues here there that should be easily fixable.

---

> ### Author Response · Authors · 2024-11-22
>
> Thank you for the encouraging review; we are gratified that you see the importance of the contribution.
>
> Regarding your question about the generality of the experiments, and whether we think they will stand the test of time on other LLMs: We believe that all autoregressive LLMs trained via teacher forcing–which is basically all the LLMs since GPT2–will have fundamental limitations on reasoning tasks (a fact that is being appreciated more widely thanks to several critical studies). In the end, verification is a reasoning task as much as generation is–and thus we are confident that our experimental results will hold on all such LLMs.
>
> While we reported all our experiments on GPT4, our partial empirical studies with other LLMs (as shown in the table below), including GPT-4o, have largely been in line with the results we reported on GPT4. Nevertheless, we are currently in the process of replicating the experiments on LLaMA, and hope to have the results incorporated before the end of the discussion period, but certainly for the camera-ready version.
>
> | Model       | Domain                | S.P | LLM+LLM |
> |-------------|-----------------------|-----|---------|
> | GPT-4o      | Graph coloring        | 9%  | 0%      |
> | GPT-4o-mini | Graph Coloring        | 0%  | 0%      |
> | GPT-4o-mini | Graph Coloring (Easy) | 30% | 0%      |
> | GPT-4o-mini | Blocksworld | 29% | 4%      |
> | GPT-4o-mini | Mystery Blocksworld | 1% | 0%      |
>
>
> > **The use of \citep and \citet is not correct. Also there are some small spelling/typesetting issues here there that should be easily fixable.**
>
> We’ve fixed the citet/citep issue, as well as updating the typesetting and fixing a few commas in the revised version. Thank you for pointing this out!

---

### Official Review · Reviewer_Dj2B · 2024-11-04

**Soundness:** 2
**Presentation:** 3
**Contribution:** 2
**Rating:** 6
**Confidence:** 5

**Summary:**

The paper evaluates the self-verification abilities of LLMs in reasoning and planning tasks using iterative prompting and critiquing. It contrasts the performance of LLMs self-verifying their solutions against external sound verifiers across three domains: Game of 24, Graph Coloring, and STRIPS planning. Findings indicate that LLMs underperform in self-verification and that external sound verification significantly improves the accuracy of solutions. The study suggests the ineffectiveness of self-critique mechanisms in LLMs and recommends integrating external verifiers for better performance in reasoning tasks.

**Strengths:**

1. The paper addresses a novel aspect of LLMs—self-critique and iterative verification—that is underexplored in the existing literature. It challenges the assumption that LLMs can effectively self-critique by demonstrating that external verification offers more reliable improvements.
2. The experimental setup is clearly described, allowing for reproducibility and understanding of how iterative prompting affects LLM performance in reasoning tasks. The paper methodically outlines its methodology and the rationale behind using specific domains for testing (Sections 4 and 3).
3. The findings significantly contribute to understanding LLM limitations in self-verification tasks, which is critical for deploying these models in real-world applications where accuracy and reliability are paramount.
4. The study is well-structured and robustly empirically analyzed, providing a comparative assessment of LLMs with and without external sound verifiers.

**Weaknesses:**

1. The paper’s focus on only three specific domains might limit the generalizability of the findings. While these domains are relevant, more varied tasks could provide a broader understanding of LLM capabilities across different reasoning types (Section 3).
2. The analysis of the self-critique mechanism lacks depth regarding why LLMs fail at self-critique. Specific instances of LLM outputs and their failures would enrich the discussion by pinpointing the flaws in LLM reasoning strategies (Section 5.1).
3. There is no detailed discussion on the computational cost and efficiency of using external verifiers versus self-verification. This information would be crucial for practical implementations where resource constraints are a significant consideration.
4. The paper does not thoroughly explore the theoretical implications of its findings on the computational complexity theories surrounding LLMs and self-verification. A deeper theoretical analysis could provide insights into the fundamental limitations of LLM architectures (Section 2).

**Questions:**

1. How do the authors anticipate their findings will generalize to other complex reasoning tasks not covered in the study? Can the observed ineffectiveness of self-critique mechanisms be extrapolated to different types of LLMs or reasoning models?
2. Could the authors elaborate on the choice of domains for the study? Why were these specific domains chosen, and how do they represent the broader spectrum of reasoning tasks applicable to LLMs?
3. What additional mechanisms or modifications do the authors suggest could potentially improve the self-verification capabilities of LLMs? Is there ongoing work to develop more effective internal critique mechanisms within LLMs?
4. How do the authors envision the impact of their findings on the future development and deployment of LLMs in safety-critical applications? What precautions or additional measures would they recommend based on their study’s outcomes?

---

> ### Author Response · Authors · 2024-11-22
>
> We thank reviewer Dj2B for their valuable comments. We are gratified that the reviewer found our work to be novel, significant and well-structured. Here are some clarifications for the concerns raised.
>
> **Choice of domains and applicability of results:**
>
> We would like to direct the reviewer to lines 139-174 and in particular to the following:
>
> > *In the current work we address this by restricting our focus to fully specified, formally verifiable problems which can be solved directly by deductive methods. Though these may at first seem like a very narrow class, especially when compared to the cornucopia of commonsense, language-based, and domain-specific benchmarks in the literature, we argue that they are fundamental, as any other reasoning tasks must include components that test these same capabilities–otherwise they are merely testing recall.*
>
> Furthermore, we specifically chose domains on which the classical argument mentioned in line 53 holds. To clarify what we mean by this: Our domains all have the property that it is easier to check potential answers than it is to generate correct ones. Formalizing this in terms of complexity theory, generation is in a higher complexity class than verification. This holds true for all our tasks, over which verification is polynomial, but generation is NP-complete or even PSPACE-complete! [1,2]
>
> What should be surprising here is that, despite this gap in the computational complexity of  verification versus generation, using the LLM as a verifier is just as brittle and inaccurate as using it just to guess solutions.
>
> In other words, previous claims that LLMs can bootstrap themselves on arbitrary problems are misleading and not robust, even when the playing field is skewed in their favor. Even worse, we've shown that the LLM self-verification loop can in some cases significantly decrease performance.
>
> Finally, we expect our conclusions to be applicable to many overall ambiguous and complex tasks, because partial verifiers can be constructed and used together. Most real-world tasks contain a mixture of formally verifiable and tacit aspects. For example, see the TravelPlanning benchmark [3] which combines “hard” and “soft” critics of LLM-generated plans. Or consider linters, unit tests, etc in software development.
>
>
>
> **What additional mechanisms or modifications do the authors suggest could potentially improve the self-verification capabilities of LLMs? Is there ongoing work to develop more effective internal critique mechanisms within LLMs?**
>
> In our view, verification/critique mechanisms can provide three things: a (potentially inaccurate) binary signal about correctness, feedback that hopefully elicits the correct answer, and guarantees over the output of the complete system. While the first two can be improved either with customized pre-training/fine tuning that is most relevant to the task, they would remain brittle and sensitive to minor distributional shifts  despite them being semantics-preserving. The third–giving guarantees on verification status–will be out of reach for pre-trained systems.
>
> While LLMs may improve on the first points over time, especially within domains that are well-represented in synthetic training data geared towards verification, their nature as uninterpretable black boxes precludes them from fulfilling the final role. Even when we intuitively think we have a grasp on how they fare on some problem class, they may in fact just be learning brittle distributional features that eventually fail in novel situations [4].
>
> **How do the authors envision the impact of their findings on the future development and deployment of LLMs in safety-critical applications? What precautions or additional measures would they recommend based on their study’s outcomes?**
>
> The self-critiquing limitations of LLMs are of particular danger in safety-critical applications–where the system might send out a potentially incorrect solution for deployment.
> As we discussed in the conclusion and introduction, we believe that verification should be offloaded to sound systems when feasible, and LLMs should not be trusted in their current form in other scenarios without significant oversight.
>
> Most complex tasks contain a mixture of explicitly verifiable and tacit aspects. We believe that partial verifiers can be constructed and used together for such problems [5]. Any sort of LLM verification must be tested thoroughly as there are cases like the ones we’ve shown where they actually worsen performance.

---

> > ### Author Response · Authors · 2024-11-22
> >
> > **Analysis of self-critique mechanism:**
> >
> > Due to page limit restrictions, we have moved our examples of specific instances and the kinds of errors that verifier LLMs make to the Appendix. We have provided a more in-depth comparison of GPT-4’s critique and verification abilities across the three domains in Appendices A3, A4 and A5. We have also provided examples of the common kinds of errors that GPT-4 makes in generating critiques. These include hallucinations about the structure of the problem (e.g. incorrectly representing preconditions, incorrectly stating vertices are connected when they aren’t), mismatches between calculations and final answer (e.g. G24 instances in which the verifying LLM states that the proposed expression simplifies to 24, but nevertheless says it is incorrect), and calculation errors (including incorrectly updating the state in planning domains).
> >
> > [1] Bylander, Tom. "The computational complexity of propositional STRIPS planning." Artificial Intelligence 69, no. 1-2 (1994): 165-204.
> >
> > [2] Aho, Alfred V., John E. Hopcroft and Jeffrey D. Ullman. “The Design and Analysis of Computer Algorithms.” (1974).
> >
> > [3] Xie, Jian, Kai Zhang, Jiangjie Chen, Tinghui Zhu, Renze Lou, Yuandong Tian, Yanghua Xiao, and Yu Su. "TravelPlanner: A Benchmark for Real-World Planning with Language Agents." In Forty-first International Conference on Machine Learning.
> >
> > [4] Zhang, Honghua, Liunian Harold Li, Tao Meng, Kai-Wei Chang, and Guy Van Den Broeck. "On the paradox of learning to reason from data." In Proceedings of the Thirty-Second International Joint Conference on Artificial Intelligence, pp. 3365-3373. 2023.
> >
> > [5] Kambhampati, Subbarao, Karthik Valmeekam, Lin Guan, Mudit Verma, Kaya Stechly, Siddhant Bhambri, Lucas Paul Saldyt, and Anil B. Murthy. "Position: LLMs Can’t Plan, But Can Help Planning in LLM-Modulo Frameworks." In Forty-first International Conference on Machine Learning.

---

### Meta-Review · Area_Chair_9GKN · 2024-12-23

**Metareview:**

In this work, the authors empirically evaluate the self-critique/self-verification methods for LLMs, which have the promise to improve LLMs' own solutions. A very interesting finding is that self-verification doesn't work well, but performance improves with sound external verification.

The reviewers found the paper well-written, novel and significant. However there are some over-claimed statements, and the scope is relatively limited to support the reliability of the conclusion. Overall the strengths outweigh the weaknesses. All reviewers consider this work above the acceptance threshold.

I recommend acceptance.

**Additional Comments On Reviewer Discussion:**

The reviewers found the paper well-written, novel and significant. However there are some over-claimed statements, and the scope is relatively limited to support the reliability of the conclusion. Overall the strengths outweigh the weaknesses. All reviewers consider this work above the acceptance threshold.

---

### Decision · Program_Chairs · 2025-01-22

Accept (Poster)